# Non-invasive measurement of mRNA decay reveals translation initiation as the major determinant of mRNA stability

Leon Y Chan[1]*, Christopher F Mugler[1†], Stephanie Heinrich[2†], Pascal Vallotton[2], Karsten Weis[2]*

[1]Department of Molecular and Cell Biology, University of California, Berkeley, Berkeley, United States; [2]Department of Biochemistry, ETH Zurich, Zurich, Switzerland

**Abstract** The cytoplasmic abundance of mRNAs is strictly controlled through a balance of production and degradation. Whereas the control of mRNA synthesis through transcription has been well characterized, less is known about the regulation of mRNA turnover, and a consensus model explaining the wide variations in mRNA decay rates remains elusive. Here, we combine non-invasive transcriptome-wide mRNA production and stability measurements with selective and acute perturbations to demonstrate that mRNA degradation is tightly coupled to the regulation of translation, and that a competition between translation initiation and mRNA decay -but not codon optimality or elongation- is the major determinant of mRNA stability in yeast. Our refined measurements also reveal a remarkably dynamic transcriptome with an average mRNA half-life of only 4.8 min - much shorter than previously thought. Furthermore, global mRNA destabilization by inhibition of translation initiation induces a dose-dependent formation of processing bodies in which mRNAs can decay over time.

DOI: https://doi.org/10.7554/eLife.32536.001

*For correspondence:
leonyenleechan@gmail.com (LYC);
karsten.weis@bc.biol.ethz.ch (KW)

†These authors contributed equally to this work

## Introduction

Gene expression is the central process that drives all other cellular processes required for life. The amounts and modification states of the mRNA and protein gene products are what ultimately determine the identity, function and fate of a given cell. The abundances of both mRNAs and proteins are in turn determined kinetically by balancing both synthetic and degradative processes. At the mRNA level, we have a detailed understanding of both how mRNAs are made and how the individual steps of transcription, splicing and maturation are regulated. However, less is known about the regulation of mRNA decay. Whereas individual steps of mRNA degradation have been determined, the question of what determines the stability of mRNAs across the transcriptome remains largely unanswered.

Bulk mRNA degradation was shown to be initiated by the removal of the polyA tail (*Shyu et al., 1991*; *Muhlrad and Parker, 1992*). This triggers degradation through one of two pathways. mRNAs can either be degraded from the 3' end by the exosome complex of 3' to 5' exonucleases or -what is thought to be more common in yeast- deadenylation is followed by removal of the 5'-methylguanosine cap by the decapping complex (*Muhlrad et al., 1994*; *Decker and Parker, 1993*). Removal of the cap structure is then followed by exonucleolytic digestion from the 5' end of the mRNA by the cytoplasmic 5' to 3' exonuclease, Xrn1. While these pathways of mRNA degradation are well elucidated, their upstream regulators remain less clear and it is not well understood how the decision is made whether an mRNA continues to be translated or enters the decay pathway.

Factors ranging from polyA tail length to mRNA structure have been proposed to affect global transcript stability but many models have been centered on how the process of translation regulates transcript lifetime. Two alternative models have been put forth to explain how mRNA decay is linked to translation (Figure 3A). The first model originates from the observation that mRNA stability significantly correlates with codon usage. It was proposed that slowly elongating ribosomes at suboptimal codons signal to the decay machinery to target the bound mRNAs for destruction. Therefore, this stalled ribosome-triggered decay model centers on the process of translation elongation (*Presnyak et al., 2015*; *Radhakrishnan et al., 2016*). The second model arises from the observations that translation and decay are inversely related and posits that bound translation factors protect an mRNA from decay. Such a translation factor-protection model predicts that translation initiation, either directly or indirectly, competes with the RNA decay machinery. In the latter model, the stability of a given transcript would be determined by a competition between the eIF4F initiation complex and the decapping complex for the 5' methylguanosine cap, and/or by ribosomes sterically blocking decay factors from the mRNA (*Beelman and Parker, 1994*; *Schwartz and Parker, 1999*; *Schwartz and Parker, 2000*; *LaGrandeur and Parker, 1999*). Both of these models have supporting experimental evidence and are also not mutually exclusive. However, the available experimental evidence for each of these models has mainly been gathered using specific reporter transcripts and methods to measure mRNA stability that can introduce unintended effects and thus might lead to non-physiological measurements of half-life as discussed below. Moreover, the perturbations that have been employed to probe the relationship between translation and decay have the potential for significant secondary effects. Thus, improved methods to both measure mRNA stability as well as perturbing core elements of the translation machinery are required to evaluate the existing models.

Classical measurements of mRNA stability for multiple transcripts in parallel have employed global inhibition of transcription. This leads to two major complications. The first is that the global inhibition of transcription is a major perturbation to the cell and this has been shown to induce a general stress response (*Sun et al., 2012*). This stress response is elicited regardless of the method of global transcription inhibition be it by pharmacological or genetic means. The second complication arises from the fact that the methods used to shutoff transcription have off-target effects themselves. Temperature shock or the use of transcriptional inhibitors such as phenantroline and thiolutin have unintended effects on cell physiology independent of transcriptional inhibition (*Sun et al., 2012*; *Harigaya and Parker, 2016*). At the individual transcript measurement level, the use of transcriptional shutoff via the regulatable *GAL* promoter has also been widely used. However, shutoff of the *GAL* promoter requires an acute carbon source shift and it has been shown that several key factors in the decay pathway such as Pat1, Dhh1, Ccr4 and Xrn1 are regulated in a carbon source-dependent manner (*Ramachandran et al., 2011*; *Braun et al., 2014*).

In this study, we have sought to determine how mRNA half-life is controlled on a transcriptome-wide level and have taken a two-pronged approach to study the relationship between translation and decay. First, we refine a metabolic labeling-based assay to measure mRNA lifetimes in a transcriptome-wide and non-invasive manner. Our new measurements reveal a much more dynamic transcriptome than previously measured by metabolic labeling with an average and median transcript half-life of only 4.8 and 3.6 min respectively. Next, we combine this measurement tool with both pharmacological and conditional genetic tools to directly perturb the processes of translation initiation and elongation. Our studies show that the competition between translation initiation and mRNA turnover determines the lifetime for an mRNA whereas slowing elongation globally leads to mRNA stabilization. At the cellular level, we find that the formation of processing bodies, sites where mRNAs are thought to be repressed and destroyed, is stimulated when translation initiation is attenuated suggesting that processing bodies form when mRNA clients are shunted into the degradation pathway.

## Results

### An improved non-invasive metabolic labeling method reveals that the yeast transcriptome is highly unstable

We and others previously demonstrated that thio-modified uracil nucleobases, such as 4-thio-uracil (4TU) are efficiently incorporated into nascent RNA (*Munchel et al., 2011*; *Miller et al., 2011*;

*Cleary et al., 2005*; *Dölken et al., 2008*). Metabolic labeling with 4TU at appropriate concentrations has no adverse effect on cell proliferation and does not lead to major changes in gene expression (*Sun et al., 2012*; *Miller et al., 2011*) (see Appendix 1). Taking advantage of the thio-reacting group, labeled RNAs can then be biotinylated allowing for an affinity-based separation of thio-labeled mRNAs from unlabeled mRNAs in pulse-labeling experiments (*Figure 1A*). When this strategy is applied in a time-resolved manner, it enables the direct measurement of mRNA decay kinetics in an unperturbed system.

We made three key modifications to our previously published protocol (*Munchel et al., 2011*). All modifications targeted the problem of inefficient subtraction of newly synthesized mRNAs, which can result from inefficient chase of a metabolic label or low enrichment for labeled RNA during biotin-mRNA separation (see Appendix 1 for a detailed discussion). First, we optimized streptavidin-bead blocking and washing conditions that significantly reduced non-specific RNA bead binding (*Figure 1A*, step 6). Notably, the extent of non-specific bead binding differed between mRNAs indicating that non-specific binding can cause transcript-specific as well as global errors in decay rate determination. Second, we switched to the recently developed MTSEA-biotin which crosslinks biotin to a free thio group with far greater efficiency than the previously used HPDP-biotin (*Duffy et al., 2015*) (*Figure 1A*, step 4). In combination, this improved efficiency allowed us to employ a 4TU)-chase labeling scheme in which rates of transcription and decay can be measured simultaneously (*Figure 1A*). Lastly, we simulated the effect that inefficiency would generally have on decay data and found that by introducing an efficiency parameter into the standard exponential decay model, we were able to account for the remaining inefficiency in labeling and capture while extracting the essential stability parameters from the data (*Appendix 1—figure 10*, *Figure 1B*). This modification of the standard exponential decay model allows one to account for the general experimental problem of inefficiency in mRNA labeling (due to uracil content and overall thio-uracil uptake), biotin crosslinking (thio-uracils that remain uncrosslinked to biotin) and biotinylated RNA separation (due to transcript-specific issues of bead and tube retention) (materials and methods).

Using this protocol, we measured half-life values for 5378 of the 6464 annotated transcripts in rapidly dividing budding yeast with a high agreement between biological replicates (Pearson correlation = 0.90) (*Figure 1—figure supplement 1C*). The analysis of the efficiency parameter showed that 92% of transcripts are labeled and separated with >90% efficiency and 98% of transcripts are labeled and separated with >80% efficiency indicating that our experimental optimizations are performing well (*Figure 1—figure supplement 1D*). Moreover, our data displayed a good fit to our modified exponential decay model with 84% of transcripts fitting with an $R^2$ of >0.95 (*Figure 1—figure supplement 1E*). Of the remaining transcripts, 209 fit the model poorly ($R^2$ <0.8) and an additional 877 could not be fit (*Figure 1—figure supplement 1B*). Most of these latter transcripts showed very low expression in our growth conditions and were thus excluded from the analysis.

The new measurements with our improved protocol revealed a much less stable transcriptome than previously reported, with average and median mRNA half-lives of 4.8 and 3.6 min respectively (*Figure 1C*). By summing the abundance of all mRNAs, we calculated the half-life of the bulk transcriptome to be 13.1 min (*Figure 1—figure supplement 1A*). Note that this value is higher than the 4.8 min average value because it takes into account transcript abundance and many of the longest-lived transcripts are present in many copies within the mRNA pool. Previously, the stability of the polyA(+) RNA pool had been measured by [14]C-adenine pulse-labeling experiments, which are the least invasive measurements that have been performed to date and could be considered the benchmark to evaluate any mRNA stability determining method. Our measurement using thio-uracil chase agrees remarkably well with [14]C-adenine pulse labeling data which reported a 11.5 min half-life for the bulk polyA(+) RNA pool in the cell (*Petersen et al., 1976*).

We also profiled the stability of the transcriptome in the absence of polyA selection by sequencing unselected, total RNAs after metabolic labeling. We found that the overall stabilities were similar: in the absence of polyA selection, the average and median mRNA half-lives were 4.9 and 4.0 min respectively compared to 4.8 and 3.6 min with polyA selection (*Figure 1—figure supplement 2A*). The correlation between half-lives measured by these two datasets was only 0.44, which is likely due to the low number of mRNA reads recovered from the total RNA reads (0.8–2.5% of total reads depending on the timepoint) when total RNA was sequenced (*Figure 1—figure supplement 2B*). Accordingly, many lower correlating transcripts were of low abundance and correlation increased amongst the higher abundance transcripts when half-lives derived from polyA selection were

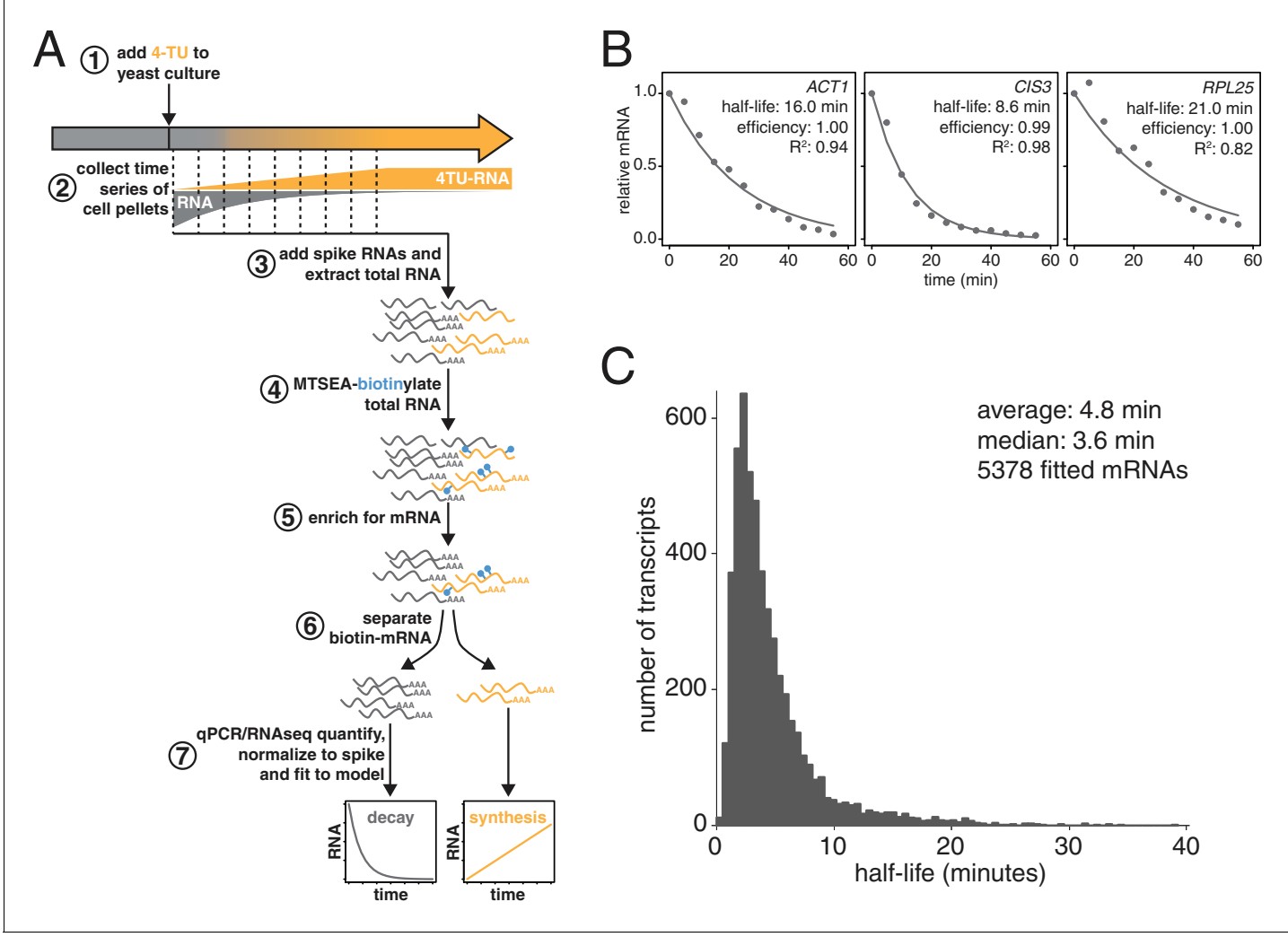

**Figure 1.** Improved mRNA stability measurements with metabolic labeling. (**A**) Experimental scheme to measure mRNA stability and production by metabolic labeling. (**B**) Wild-type cells (KWY165) were subjected to the experiment described in (**A**) and decay rates for *ACT1*, *CIS3* and *RPL25* transcripts were determined by RT-qPCR. (**C**) mRNA samples in (**B**) were quantified by RNA-seq to determine mRNA stabilities across the transcriptome. Half-lives are plotted by frequency and each bin is 0.5 min wide.

DOI: https://doi.org/10.7554/eLife.32536.002

The following source data and figure supplements are available for figure 1:

**Source data 1.** Source data for *Figure 1B*: decay kinetics of ACT1, CIS3 and RPL25 mRNAs.

DOI: https://doi.org/10.7554/eLife.32536.005

**Source data 2.** Source data for *Figure 1C* and *Figure 1—figure supplement 1C*: transcriptome wide decay data for two biological replicates of wild-type yeast cells grown in exponential phase.

DOI: https://doi.org/10.7554/eLife.32536.006

**Figure supplement 1.** Characterization of transcriptome-wide mRNA stability profiling.

DOI: https://doi.org/10.7554/eLife.32536.003

**Figure supplement 2.** Comparison of mRNA stabilities in the presence or absence of mRNA enrichment.

DOI: https://doi.org/10.7554/eLife.32536.004

**Figure supplement 2—source data 1.** Source data for *Figure 1—figure supplement 2A–B*: transcriptome wide decay data prepared without polyA selection.

DOI: https://doi.org/10.7554/eLife.32536.007

compared to unselected RNA. (*Figure 1—figure supplement 2C*). However, for specific transcripts, biological differences in mRNA decay steps downstream of deadenylation such as decapping and exo-nucleolytic processing probably also contribute to the differences between the two

measurements. Nevertheless, we conclude that the overall stability of the transcriptome remains largely unchanged in the absence of polyA selection indicating that for the majority of transcripts, deadenylation is the rate determining step for decay.

Consistent with the extensive protocol optimization, we found an overall poor correlation with our previously published dataset (*Figure 1—figure supplement 1F*). Nonetheless, our current measurements are consistent with the findings of Munchel et al. that long-lived (>1 SD above the mean) transcripts are functionally enriched for translation factors and that ribosomal protein-encoding mRNAs specifically are long lived as a group with an average half-life of 15.5 min (*Figure 1—figure supplement 1.G*; *Figure 1—figure supplement 1.H*) (*Munchel et al., 2011*). There is no significant functional enrichment in genes with exceptionally short (<1 SD below the mean) mRNA half-lives. Our dataset does not agree well with the datasets derived from global transcriptional inhibition, which cluster with each other (*Harigaya and Parker, 2016*)(*Figure 1—figure supplement 1I*). This is consistent with the findings of Sun et al. and Harigaya et al. that methods relying on transcriptional inhibition all induce a global stress response that is elicited regardless of the method of transcriptional inhibition (*Sun et al., 2012*; *Harigaya and Parker, 2016*). Instead, our dataset clusters with the datasets of Cramer and Gresham that also employed non-invasive metabolic labeling although the transcriptome is much less stable by our measurements (*Figure 1—figure supplement 1I*) (*Sun et al., 2012*; *Miller et al., 2011*; *Neymotin et al., 2014*). This shorter half-life of the transcriptome is likely due to improvements in biotin-crosslinker technology, the inclusion of multiple time-points in rate determination, cleaner methods for separating biotinylated-RNAs from unlabeled RNAs as well as improvements to the modeling and extraction of half-life parameters from the decay measurements. The overall distribution of half-lives for all fitted mRNAs (*Figure 1C*) is non-Gaussian stretching across more than an order of magnitude. The shortest half-lives are less than 1 min whereas the most stable transcripts have half-lives of more than 30 min.

## Slowing translation elongation protects transcripts against degradation

To begin to identify factors that regulate this half-life diversity, we compared our decay dataset to other transcriptome-wide datasets of various mRNA measurements (*Figure 2*). Our decay data clustered with transcript abundance, metrics of codon usage (normalized translational efficiency (nTE) and codon adaptation index (CAI)), as well as translational efficiency measured by ribosome footprinting (*Pechmann and Frydman, 2013*; *Drummond et al., 2006*). The positive relationship between abundance and half-life supports the notion that mRNA levels are not only primarily dictated by the rate of synthesis, but that differential mRNA stability contributes to the regulation of transcript abundance as well. Interestingly, mRNA half-life was negatively correlated with polyA-tail length consistent with prior observations (see discussion) (*Subtelny et al., 2014*).

Our correlation analyses support prior work pointing to mRNA translation efficiency as a critical determinant of mRNA half-life. The aforementioned stalled ribosome-triggered decay and translation factor-protection models attempt to explain the positive correlations between mRNA half-life and codon usage and mRNA half-life and translation efficiency respectively (*Figure 3A*). These two models make clear and opposing predictions for how perturbing the processes of translation elongation or initiation impacts transcript stability. The stalled ribosome-triggered decay model predicts that mRNAs are destabilized upon slowing elongation whereas the translation factor-protection model predicts the opposite since slowly elongating ribosomes would accumulate on a given transcript and thus provide greater steric exclusion of decay factors. In contrast, when translation initiation rates are attenuated, the stalled ribosome-triggered decay model predicts that transcripts would either have the same stability or possibly even increased stability as once the bound ribosomes complete translation, the naked mRNA would be freed from decay-triggering ribosomes. The translation factor-protection model again predicts the opposite outcome: decreasing the rate at which translation is initiated leaves the 5' cap more exposed to the decapping machinery and fewer loaded ribosomes allows the decay factors greater access to the transcript culminating in an overall decrease in transcript stability.

To begin to test these predictions, we directly perturbed the process of translation elongation and observed the effects on mRNA decay. Consistent with previous reports, we found that treating cells with a strong dose (50 µg/mL) of the elongation inhibitor cycloheximide resulted in stabilized *CIS3* and *RPL25* transcripts and had no significant effect on *ACT1* mRNA stability (*Figure 3—figure supplement 1A*) (*Beelman and Parker, 1994*). A potentially problematic aspect of these

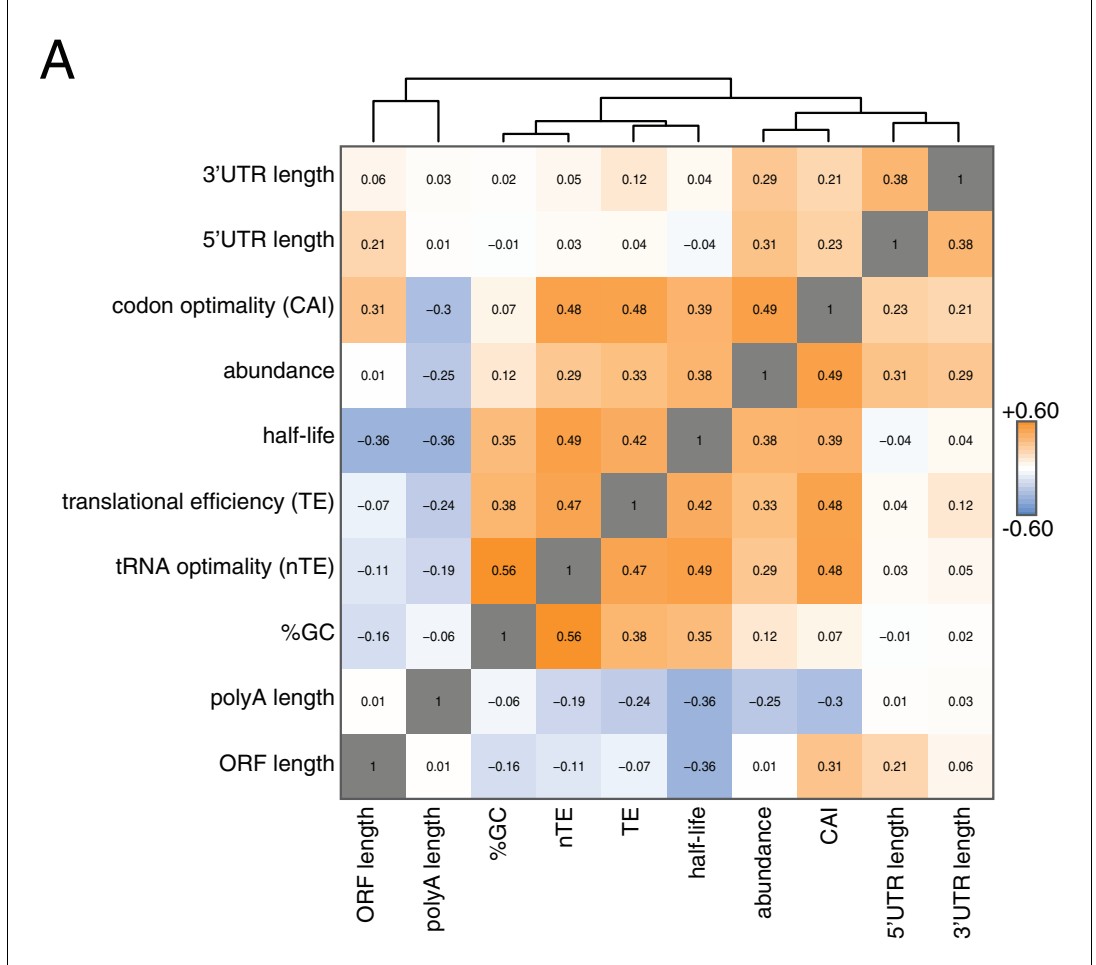

**Figure 2.** Correlation of mRNA features. (**A**) Spearman rank correlation coefficients were computed for pairs of mRNA parameters of stability (half-life), translation efficiency (TE), polyA tail length, codon optimality (CAI), tRNA optimality (nTE), abundance, UTR lengths, GC content and ORF length and plotted as a heatmap. Datasets were hierarchically clustered based on Euclidian distances. Orange represents positive correlation and blue represents negative correlation. Correlations between identical datasets are colored in gray. See **Supplementary file 1** for sources of genome wide data.

DOI: https://doi.org/10.7554/eLife.32536.008

The following figure supplement is available for figure 2:

**Figure supplement 1.** Correlation between translation initiation rate and aspects of mRNA structure and function.

DOI: https://doi.org/10.7554/eLife.32536.009

experiments is that high doses of cycloheximide completely halt translation thus shutting down a myriad of cellular processes, which in turn could lead to indirect effects. To address this issue, we titrated the amount of cycloheximide to a sub-lethal concentration (0.2 μg/mL) and assayed the effect of this level of elongation inhibition on transcript stability (**Figure 3—figure supplement 2H**). Even with this low concentration of cycloheximide, the *CIS3* and *RPL25* transcripts remained stabilized and the *ACT1* mRNA again was not significantly affected (**Figure 3B**). The effects of cycloheximide on mRNA turnover were specific to elongation inhibition as mRNA half-lives were unchanged in a cycloheximide resistant mutant (*rpl28-Q38K*) (**Figure 3—figure supplement 1B**). We next tested an alternative translation elongation inhibitor, sordarin, which blocks the function of eukaryotic elongation factor 2 (**Shastry et al., 2001**). When cells were treated with a sub-lethal dose of sordarin, we again observed a stabilizing effect on mRNA half-lives (**Figure 3C** and **Figure 3—figure supplement 1C**). These stabilization effects were not due to an inability to incorporate 4TU into newly made transcripts as mRNA synthesis rates were not reduced upon treatment with either cycloheximide or sordarin (**Figure 3—figure supplement 1D and E**). Moreover, these effects were not specific to polyA selection. When these experiments were analyzed using total RNA, mRNAs were stabilized upon

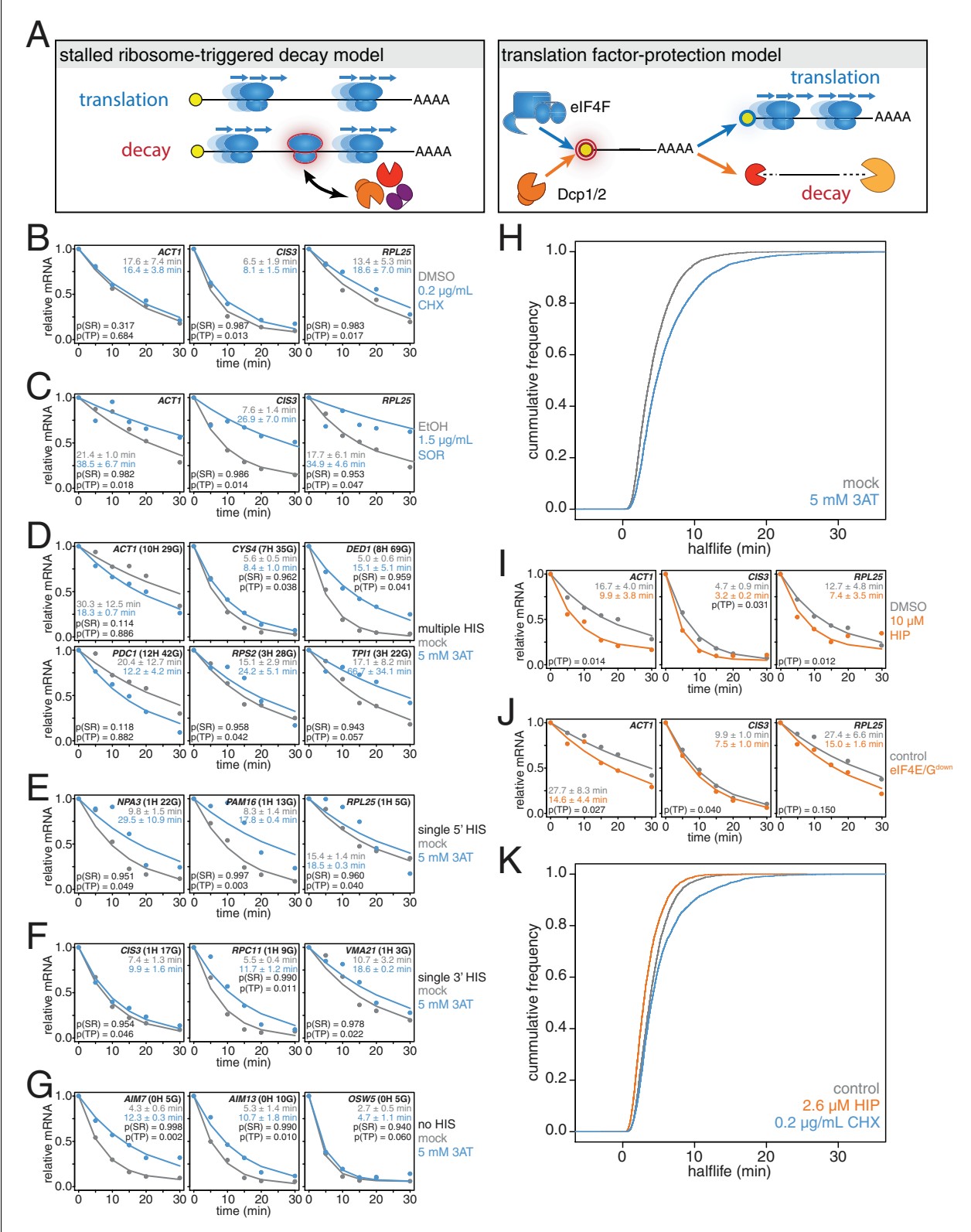

**Figure 3.** mRNAs are stabilized by slowly elongating ribosomes and destabilized when translation initiation is inhibited. (**A**) Cartoon depictions of the stalled ribosome-triggered decay and translation factor-protection models. (**B**) Wild-type cells (KWY165) were subjected to mRNA stability profiling immediately after addition of 0.1% DMSO or 0.2 μg/mL cycloheximide in 0.1% DMSO. Data on *ACT1*, *CIS3* and *RPL25* mRNAs were collected and plotted. See *Figure 3—figure supplement 4A* for biological replicates. P-values are computed using a one-sided paired t-test for both the stalled

*Figure 3 continued on next page*

*Figure 3 continued*

ribosome-triggered decay model (p(SR)) as well as the translation factor-protection model (p(TP)). P-values less than 0.05 are significant. (**C**) Wild-type cells (KWY165) were subjected to mRNA stability profiling 33 min after addition of 0.1% ethanol or 1.5 µg/mL sordarin in 0.1% ethanol (note that this is the timepoint when a growth defect is manifested, see *Figure 3—figure supplement 1C*). Data were collected, analyzed and plotted as in *Figure 3B*. See *Figure 3—figure supplement 4B* for biological replicates. (**D–G**) *HIS3 gcn2Δ* cells (KWY7337) were subjected to mRNA stability profiling immediately after non-addition (mock) or addition of 5 mM 3AT. Data were collected, analyzed and plotted as in *Figure 3B*. See *Figure 3—figure supplement 4C* for biological replicates. (**H**) mRNA samples collected from the experiment described in *Figure 3D–G* were subjected to global mRNA stability profiling. Cumulative frequencies of transcript half-life are plotted. (**I**) Wild-type cells (KWY165) were subjected to mRNA stability profiling immediately after addition of 0.1% DMSO or 10 µM hippuristanol. Data were collected, analyzed and plotted as in *Figure 3B*. p-values were not computed for the stalled ribosome-triggered decay model as this model does not make a clear prediction as to how mRNA stability is affected when translation initiation is perturbed. See *Figure 3—figure supplement 5A* for biological replicates. (**J**) *pGPD1-LexA-EBD-B112 CDC33-3V5-IAA7* pRS425 cells (KWY7336: control) and *pGPD1-LexA-EBD-B112 CDC33-3V5-IAA7 pGPD1-OsTIR1* pRS425-*p4xLexOcyc1-CDC33ΔCAP* cells (KWY7334: eIF4E/G$^{down}$) were grown in CSM-LEU-0.5xURA pH5.5 media and subjected to mRNA stability profiling immediately after addition of 10 nM β-estradiol, 100 µM 3-indoleacetic acid and 4 µM IP$_6$. Data were collected, analyzed and plotted as in *Figure 3I*. See *Figure 3—figure supplement 5B* for biological replicates. (**K**) Wild-type cells (KWY165) were subjected to global mRNA stability profiling immediately after addition of 0.1% DMSO (gray) or 2.6 µM hippuristanol (orange) or 0.2 µg/mL cycloheximide (blue). Cumulative frequencies of transcript half-life are plotted.
DOI: https://doi.org/10.7554/eLife.32536.010

The following source data and figure supplements are available for figure 3:

**Source data 1.** Source data for *Figure 3B, C, D–G, I and J*: decay kinetics of selected mRNAs in cells treated with translational inhibitors.
DOI: https://doi.org/10.7554/eLife.32536.014
**Source data 2.** Source data for *Figure 3H*: Transcriptome wide decay data for cells treated or mock treated with 3AT.
DOI: https://doi.org/10.7554/eLife.32536.015
**Source data 3.** Source data for *Figure 3K*: Transcriptome wide decay data for cells treated with DMSO, cycloheximide or hippuristanol.
DOI: https://doi.org/10.7554/eLife.32536.016
**Figure supplement 1.** Effects of translation elongation inhibitors on mRNA stability and cell growth.
DOI: https://doi.org/10.7554/eLife.32536.011
**Figure supplement 2.** Biological replicates of experiments described in *Figure 3B–G*.
DOI: https://doi.org/10.7554/eLife.32536.012
**Figure supplement 3.** Effects of different modes of translation inhibition on mRNA stability in the absence of mRNA enrichment.
DOI: https://doi.org/10.7554/eLife.32536.013
**Figure supplement 4.** Biological replicates of experiments described in *Figure 3I–J*.
DOI: https://doi.org/10.7554/eLife.32536.017
**Figure supplement 5.** Characterization of conditional mutants of translation initiation.
DOI: https://doi.org/10.7554/eLife.32536.018

elongation inhibition and the half-life differences were unchanged in the case of low cycloheximide or further exaggerated as in the cases of high cycloheximide and sordarin (*Figure 3—figure supplement 3A–C*). The dramatic increase in mRNA stability in sordarin or high doses of cycloheximide is consistent with previous findings that decapping rather than deadenylation is blocked upon cycloheximide treatment (*Figure 3—figure supplement 3A and C*)(*Beelman and Parker, 1994*). Using a one-sided paired t-test, we find that 5 of the six measurements support the translation factor protection model (p(TP)<0.05) and none of the measurements support the stalled ribosome protection model (p(SR)<0.05). We conclude that inhibiting translation elongation stabilizes mRNAs.

While these results demonstrate that a stalled ribosome per se is not sufficient to induce decay, we could not exclude that cycloheximide or sordarin treatment might only poorly imitate slowed ribosomes on non-optimal codons since the acceptor-site of the ribosome remains occupied when these drugs are employed (*Roy and Jacobson, 2013*). To best mimic a non-optimal codon where the acceptor-site would be unoccupied, we treated cells with a sub-lethal dose (5 mM) of 3-amino-1,2,4-triazole (3AT), which results in histidine starvation thus lowering the concentration of histidyl-tRNAs (*Figure 3—figure supplement 1F*) (*Klopotowski and Wiater, 1965*). Indeed 3AT has previously been shown to stall ribosomes at histidine codons (*Guydosh and Green, 2014*). Histidine starvation also affects translation initiation by phosphorylation of eukaryotic initiation factor 2α via the Gcn2 kinase (*Hinnebusch, 2005*). In order to examine the effect on translation elongation by 3AT in isolation, all 3AT experiments were thus performed in *gcn2Δ* mutant cells. We examined the stability of 15 mRNAs with diverse spacing and position of histidine codons either untreated or treated with 3AT. Of these 15 paired measurements, we found that 11 are significantly stabilized and support the

translation factor protection model (p(TP)<0.05). We find that none of the measurements support the stalled ribosome triggered decay model (p(SR)<0.05) (*Figure 3D–G*). This overall stabilization effect could not be explained by poor 4TU uptake as mRNA synthesis rates were not reduced upon 3AT treatment (*Figure 3—figure supplement 1G*). Interestingly, transcripts lacking histidine codons were also stabilized, which is consistent with the observation that 3AT limits glycine availability in addition to the depletion of histidine (*Vital-Lopez et al., 2013*). Again, we found that these results were recapitulated in the absence of polyA selection suggesting that the effect of 3AT is not limited to deadenylation but apply to steps downstream in mRNA decay as well (*Figure 3—figure supplement 3D*).

To understand the effect of 3AT on transcript stability at a global scale, we also subjected our samples to transcriptome-wide stability profiling. The stabilizing effects of 3AT that we observed at the single transcript level were also seen in a highly significant manner across the transcriptome (two-sided Wilcoxian paired test: p=1.165e-199) (*Figure 3H*). The impact of histidine and glycine content appears to display as a threshold effect and once a critical number of these amino acids was reached (greater than 2), a larger fold increase in half-life was observed (*Figure 3—figure supplement 1H*). Importantly, these experiments are in line with experiments from the Jacobson group that employed conditional alleles of genes involved in tRNA maturation or aminoacylation and observed stabilization of transcripts when charged tRNA levels were reduced (*Peltz et al., 1992*; *Mangus and Jacobson, 1999*). Altogether, we conclude that inhibiting translation elongation by three different methods led to an overall stabilization of mRNAs. These observations are not consistent with the predictions of the stalled ribosome-triggered decay model but instead support a translation factor-protection model.

## Inhibition of translation initiation destabilizes individual transcripts

We next studied the effects of inhibiting translation initiation on mRNA decay. We first made use of hippuristanol, an inhibitor of eukaryotic initiation factor 4A (eIF4A) (*Bordeleau et al., 2006*). We observed that *ACT1*, *CIS3* and *RPL25* mRNAs all decayed with faster kinetics when eIF4A was inhibited (*Figure 3I*). We also attempted to generate hippuristanol-resistant alleles of the eIF4A encoding genes, *TIF1* and *TIF2*, to test the specificity of hippuristanol, but these mutations (V326I, Q327G and G351T) led to severe cell sickness (data not shown) (*Lindqvist et al., 2008*). To exclude any potential indirect effects of hippuristanol, we sought alternative means to inhibit translation initiation. Overexpression of a 5'cap-binding mutant of eIF4E (*cdc33-W104F-E105A* henceforth *cdc33*$^{\Delta CAP}$) using a β-estradiol inducible promoter caused a subtle inhibition of growth (*Marcotrigiano et al., 1997*; *Ottoz et al., 2014*) (*Figure 3—figure supplement 2B*). This defect was fully suppressed by introducing in cis the Δ1–35 (henceforth *cdc33*$^{\Delta G}$) mutation that abolishes eIF4G binding indicating that overexpression of *cdc33*$^{\Delta cap}$ leads to a dominant-negative loss of eIF4G function likely through a sequestration mechanism (*Figure 3—figure supplement 2A C*) (*Gross et al., 2003*). In addition, we placed eIF4E under control of an auxin-inducible degron system (*CDC33-3V5-IAA7*) (*Nishimura et al., 2009*). This approach alone led to a mild growth defect upon the addition of auxin presumably because eIF4E could not be fully depleted (*Figure 3—figure supplement 2D–F*). However, when these two strategies were combined to simultaneously downregulate eIF4E and eIF4G function, we observed a strong synthetic growth defect (*Figure 3—figure supplement 2G*). This system thus enabled us to acutely inhibit initiation in a manner orthogonal to hippuristanol and evaluate the resulting effects on mRNA decay. As with hippuristanol-treated cells, we found that *ACT1* and *CIS3* transcripts were significantly destabilized while the *RPL25* transcript was not significantly affected when translation initiation is slowed (*Figure 3J*). This effect was independent of polyA selection, and as for our experiments where we slowed translation elongation, we obtained comparable results when a polyA selection step was omitted (*Figure 3—figure supplement 3E–F*). Based on the results of two independent experimental approaches we conclude that inhibiting translation initiation leads to accelerated mRNA decay.

## Translation elongation and initiation globally affect mRNA half-lives

To test the generality of our findings, we also performed transcriptome-wide mRNA stability profiling of cells treated with either cycloheximide or hippuristanol. To allow for a meaningful comparison, we used hippuristanol at a sub-lethal concentration that confers a near identical growth defect as

our sub-lethal concentration of cycloheximide (*Figure 3—figure supplement 2H*). In support of our observations with individual mRNAs, cycloheximide induced a global stabilization of mRNAs (p=6.298e-106 two-sided Wilcoxon paired test) whereas hippuristanol treatment led to shorter mRNA half-lives (p=1.864e-260 two-sided Wilcoxon paired test) (*Figure 3K*). Importantly, the Spearman rank correlation coefficient between these datasets was high ($R_{sp}$(DMSO:HIP)=0.81 and $R_{sp}$(DMSO:CHX)=0.79). This suggests that these drugs did not result in a reordering of the stability profile of the transcriptome or differentially affect specific classes of mRNAs. Instead, this indicates that the drugs generally shifted the profile towards more (cycloheximide) or less (hippuristanol) stable. We conclude that slowing initiation accelerates mRNA turnover while inhibiting elongation slows mRNA turnover and that on a transcriptome-wide level, the efficiency of initiation either directly through 5'-cap competition or indirectly through ribosome protection is a major determinant of transcript stability.

## Inhibition of translation initiation induces processing bodies

What are the consequences of these perturbations to translation and their effect on mRNA decay at the cellular level? Inhibition of elongation with cycloheximide was previously shown to inhibit the formation of processing bodies (PBs), which are thought to be sites of transcript repression and decay (*Sheth and Parker, 2003*; *Kroschwald et al., 2015*; *Mugler et al., 2016*). To test the effects of inhibiting translation initiation on PB formation, we treated cells expressing Dhh1-GFP and Dcp2-mCherry markers of PBs with a range of hippuristanol concentrations. Interestingly, hippuristanol induced PB formation in a concentration dependent manner: at high doses (10–40 µM), rapid and robust PB formation could be observed; at an intermediate dose (5 µM), PBs formed over time and at a low dose (2.5 µM), PBs could not be detected (*Figure 4A and B*). These observations are consistent with previous reports showing that mutations in eIF3b enhanced PB formation(*Teixeira et al., 2005*; *Brengues et al., 2005*). Our results show that hippuristanol generates client mRNAs for the decay machinery through its inhibition of initiation. The observed dosage effect therefore suggests that PB formation is directly dependent on the number of mRNA substrates available for degradation and that microscopic PBs can only be detected when a certain threshold of decay targets is reached. Consistent with such a model, we observed the rapid relocalization of three distinct mRNAs, *GFA1*, *PGK1* and *FBA1,* to PBs upon hippuristanol-induced PB formation (*Figure 4D*). Unlike in mammalian cell culture systems, hippuristanol does not trigger the formation of stress granules in yeast (*Figure 4—figure supplement 1A*) but as with other PBs, the formation of hippuristanol-induced Dhh1- and Dcp2-containing foci requires the RNA and ATP binding activities of Dhh1 as mutants of Dhh1 that are unable to bind RNA ($dhh1^{3x-RNA}$) or ATP ($dhh1^{Q-motif}$) do not form PBs upon hippuristanol treatment (*Figure 4—figure supplement 1B–C*) (*Mugler et al., 2016*; *Mazroui et al., 2006*). An alternate explanation for these hippuristanol-induced PBs is that the perturbation of translation alone may result in cellular stress and PB formation. However, co-treatment of hippuristanol-treated cells with either cycloheximide or sordarin suppressed PB formation, suggesting that the increased number of ribosome-unbound mRNA clients available for degradation, rather than crippled translation, was causative for PB formation (*Figure 4A and C*).

Recent evidence has supported the notion that mRNAs can be degraded in PBs (*Mugler et al., 2016*; *Heinrich et al., 2017*). To examine whether we can observe mRNA degradation in PBs that form upon addition of hippuristanol, we placed a model transcript (3xGST) containing PP7 stem loops (PP7sl), which have previously been shown to be slowly decayed, under control of a β-estradiol inducible promoter (*Heinrich et al., 2017*). We pulsed cells with this transcript by treating the cells for 40 min with β-estradiol, washed out the inducer, immediately added 40 µM hippuristanol and then observed the localization of the PP7 stem loops over time. As observed for endogenous mRNAs, we found that the PP7sl-containing transcript rapidly localized to PBs (*Figure 4E*). Moreover, we found that the PP7-mRNA signal decayed over time within the PB (*Figure 4E and F*). This suggests that mRNAs localize to PBs when initiation is inhibited and that these mRNAs can be degraded after they localize to a PB. In combination with our metabolic labeling studies, we further conclude that inhibiting translation initiation leads to global mRNA destabilization which in turn causes the formation of PBs. In the presence of agents that inhibit translation elongation, mRNAs become stabilized reducing the flux of new client mRNAs into the degradation pathway, which in turn suppresses the formation of PBs.

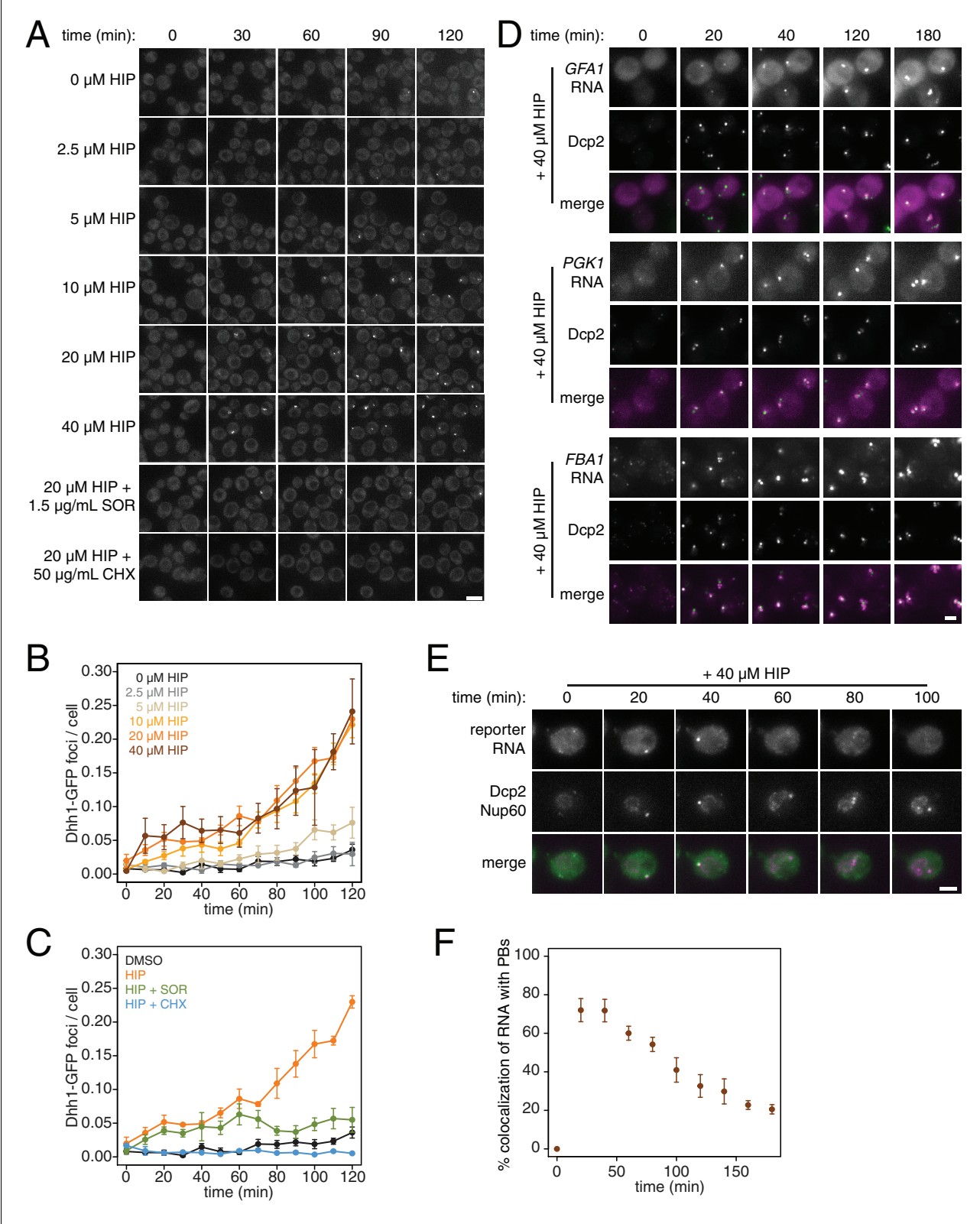

**Figure 4.** PB formation is stimulated by inhibiting translation initiation and blocked when translation elongation is inhibited. (**A**) Dhh1-GFP, Dcp2-mCherry expressing cells (KWY5948) were grown to exponential phase and then treated with 0.1% DMSO, the indicated concentration of hippuristanol or co-treated with the indicated concentration of hippuristanol and either sordarin or cycloheximide. Images were acquired every 5 min using a wide-field microscope and the images were deconvolved. Shown are maximum projections of 8 z-stacks at a distance of 0.4 µm apart. Scale bar: 5 µm. (**B–C**)
*Figure 4 continued on next page*

*Figure 4 continued*

Number of Dhh1-GFP foci per cell from experiment in (**A**) was counted using Diatrack 2.5 particle tracking software. Error bars represent SEM (n = 3 biological replicates,>300 cells counted per experiment). (**D**) Dcp2-GFP, PP7CP-mKate2 expressing cells carrying PP7sl tagged copies of *GFA1* (KWY7246), *PGK1* (KWY6963) or *FBA1* (KWY7245) were treated with 40 µM hippuristanol and immediately imaged. Images where acquired every 20 min using a wide-field microscope. Shown are maximum projections of 8 z-stacks at a distance of 0.5 µm apart. Scale bar: 2 µm. (**E**) Dcp2-mCherry, Nup60-3xmKate2, PP7CP-GFP expressing cells carrying a synthetic 3xGST-24xPP7sl under β-estradiol inducible control (KWY7227) were grown to mid-exponential phase, treated with 400 nM β-estradiol for 40 min and then transferred to media lacking β-estradiol and containing 40 µM hippuristanol and immediately imaged (see *Figure 4—figure supplement 1D* for the no hippuristanol control). Images were acquired every 20 min using a wild-field microscope. Shown are maximum projections of 8 z-stacks at a distance of 0.5 µm apart. Scale bar: 5 µm. For DMSO control images, see *Figure 4—figure supplement 1D*. (**F**) Images acquired in (**E**) were quantified for the colocalization of PP7CP-GFP foci with Dcp2-mCherry foci using FIJI software. Error bars represent SEM (n = 4 biological replicates,>120 PBs counted per timepoint).

DOI: https://doi.org/10.7554/eLife.32536.019

The following source data and figure supplement are available for figure 4:

**Source data 1.** Source data for *Figure 4B, C and F*: accumulation kinetics of P-bodies and decay of RNA in P-bodies in cells treated with translational inhibitors.
DOI: https://doi.org/10.7554/eLife.32536.021

**Figure supplement 1.** Characterization of P-body and stress-granule behavior in response to translation initiation inhibition.
DOI: https://doi.org/10.7554/eLife.32536.020

## Discussion

In this work, we have refined an assay to measure the kinetics of mRNA synthesis and decay based on 4TU metabolic labeling. This approach and similar approaches supersede the traditional methods of transcriptional inhibition as they enable quantitative and global measurements of mRNA kinetics in physiologically unperturbed cells. We used this approach to address the important question of how the process of translation affects transcript stability. Importantly, all of the measurements and experimental perturbations employed here relied on minimally invasive and rapidly inducible methods. Moreover, the drugs we used have specific molecular targets and the genetic inhibitions of eIF4G and eIF4E are induced by hormones from orthologous systems, which have minimal off-target effects.

Taken together, these approaches have enabled us to identify translation initiation as the central hub in globally regulating mRNA stability. While no direct measurement of translation initiation rates at the transcriptome-wide level has been reported, rates estimated by using a kinetic model of ribosome flux (*Ciandrini et al., 2013*) are in agreement with our conclusion and the dataset of estimated rates strongly correlates and clusters most closely with transcript half-life (*Figure 2—figure supplement 1A–B*). While we cannot exclude that suboptimal codons enhance the decay of specific transcripts, our results do not support the hypothesis that stalled ribosomes are the primary regulator of cellular mRNA decay on a transcriptome-wide level. Strong evidence for a decay mechanism triggered by sub-optimal codons was derived from experiments employing model transcripts with engineered, de-optimized codon compositions upon transcriptional shut-off (*Radhakrishnan et al., 2016*). These experiments are important but they relied on transcriptional inhibition or repression, which could be problematic as the cellular physiology is affected in such experiments and thus conclusions as to the stability of the transcripts might only be applicable to these stress conditions. More generally, the use of steady state levels to infer mRNA stabilities of engineered model transcripts has been used to argue for a role of codon usage in mRNA stability determination (*Hoekema et al., 1987*; *Boël et al., 2016*). These analyses are complicated by the fact that GC content alone affects the synthesis rate of mRNAs, and thus it might be necessary to measure both synthesis and decay parameters directly (*Newman et al., 2016*; *Zhou et al., 2016*). It could also be important to consider that engineered transcripts containing high percentages of non-optimal codons, often at levels that are not found within endogenous transcripts, might be disposed of by alternate decay pathways including quality control mechanisms such as no-go decay (*Shoemaker and Green, 2012*). Nevertheless, as previously reported, we also found a positive correlation between codon optimality and transcript stability (*Presnyak et al., 2015*). However, similar positive correlations can also be identified with general translation efficiency and transcript abundance. We therefore consider it likely that these properties have undergone similar selection

pressure and might have co-evolved with transcript stability to ensure optimal gene expression of particular highly abundant transcripts.

It has been previously proposed that deadenylation is the rate limiting step of mRNA decay (*Breunig et al., 1993*). The observation that mRNA half-lives positively correlate when measured using polyA selection compared to measurements in the absence of polyA enrichment support this model (*Figure 1—figure supplement 2B*). If the rate of deadenylation for each transcript was constant, one would therefore expect that the length of the polyA tail would directly determine the stability of the associated transcript. However, rather than positively correlating with half-life, we find that polyA tail length negatively correlates with transcript stability consistent with prior results (*Subtelny et al., 2014*). Despite this inverse relationship, it is important to note the negative effects of polyA-binding protein on transcript decapping and thus the roles of deadenylation and the length of the polyA tail in controlling transcript stability are likely more nuanced than a simple rate-limiting model would imply (*Caponigro and Parker, 1995*; *Wilusz et al., 2001*). Moreover, it will be important to examine not only a snapshot of the steady state polyA tail length but to determine the kinetics of polyA tail shortening to understand if and how the rate of deadenylation contributes to overall transcript stability.

Our work also suggests that a sudden increase of decay clients leads to PB formation once a critical threshold is reached. This is consistent with previous studies showing that mRNA is required for PB formation and further implies that mRNA can be limiting for PB formation when translation is rapidly down-regulated as is the case during cellular stress. Furthermore, as mRNA decay and translation are opposing fates for an mRNA and are competing processes in the cell, it might also be the case that the cell physically compartmentalizes these processes away from one another by use of a liquid-liquid phase transition droplet such as a P-body. A remaining open question is whether PBs form because the decay machinery is overburdened and decay intermediates accumulate or whether decay substrates are delivered to PBs in order to accelerate their decay. Inducing PBs by hippuristanol treatment will be an excellent approach to dissect these possibilities as this mode of PB formation bypasses the typical stresses associated with PB formation such as nutrient starvation that might have more pervasive and confounding effects on mRNA decay itself. The role of PBs in mRNA turnover has remained unclear and controversial. Our data support the notion that mRNAs can be degraded after being delivered to a PBs. Yet, it has also been shown that mRNAs can decay co-translationally (*Hu et al., 2009*). However, given that large quantities of mRNA needed to be purified to detect co-translational mRNA decay and that mRNA decay intermediates can only be visualized in PBs in the presence of mRNA stabilizing mutations or cis-stabilizing structures, it seems likely that neither of these modes of mRNA decay represent the primary pathways by which most mRNAs are destroyed (*Heinrich et al., 2017*; *Pelechano et al., 2015*; *Carroll et al., 2011*). We therefore favor a model in which most mRNAs are decayed in mRNPs that have exited translation and are composed of deadenylation, decapping and exo-nucleolytic factors existing apart from microscopically visible PBs (*Teixeira and Parker, 2007*).

A revelation from this work is the overall short half-life of the transcriptome, only 4.8 min or a mean lifetime of 6.9 min. This value is three times faster than was previously measured by metabolic labeling and up to 26 times faster than what was measured by transcriptional inhibition. Despite these very short half-lives, with an estimated average translation initiation rate of 0.12 s$^{-1}$, this implies that the average transcript can still code for about 50 polypeptides before it is destroyed (*Ciandrini et al., 2013*). This overall instability of the transcriptome argues against the need for regulated mRNA decay for the bulk of transcripts in the cell. That being said, there is a class of long lived transcripts that we and others have found to be enriched for translation factors and ribosomal protein encoding mRNAs, and there is indeed mounting evidence that these transcripts can have dramatically differing stabilities depending on the state of the cell (*Bregman et al., 2011*; *Gupta et al., 2016*). It is also important to note that our measurements were made in rapidly dividing yeast cells, and it remains to be examined whether the determinants of mRNA stability as well as the degree of regulated turnover could shift as cells are exposed to stresses or undergo differentiation programs. Our non-invasive metabolic labeling approach can be applied in such contexts to determine how decay and synthesis work together to kinetically shape dynamic gene expression programs.

## Materials and methods

### Yeast Strains and Growth Conditions

All strains are derivatives of W303 (KWY165) with the following exceptions: KWY7227 and KWY7246 are derivatives of BY4741 (KWY1601). All strains are listed in *Supplementary file 2*. *gcn2Δ*, *CDC33-IAA7-3V5*, *FBA1-PP7sl*, *GFA1-PP7sl* and *PGK1-PP7sl* were generated by standard PCR based methods (*Longtine et al., 1998*). *RPL28(Q38K)* was generated by plating wild-type cells on 3 mg/mL cycloheximide plates, selecting for suppressors, backcrossing the suppressors at least three times and confirming the mutation by sequencing. *HIS3* was generated by PCR replacement of the *his3-11,15* allele using *HIS3* from pRS303. *leu2-3,112Δ::CG-LEU2::pGPD1-OsTIR1*, *his3-11,15Δ::CG-HIS3::pGPD1-OsTIR1*, *trp1-1Δ::CG-TRP1::pGPD1-LexA-EBD-B112*, *his3-11,15Δ::CG-HIS3::pGPD1-LexA-EBD-B112* and *SCO2::p4xLexOcyc1-3xGST-V5-24xPP7sl-tCYC1-NatNT2* were generated by transforming strains with plasmids pKW2830 (PmeI digested), pKW2874 (PmeI digested), pKW3908 (SwaI digested), pKW4073 (SwaI digested) and pKW4190 (NotI/AscI digested) respectively. Strains were grown in CSM-lowURA (7 g/L YNB, 2% dextrose, 20 mg/L adenine, 20 mg/L arginine, 20 mg/L histidine, 60 mg/L leucine, 30 mg/L lysine, 20 mg/L methionine, 50 mg/L phenylalanine, 200 mg/L threonine, 20 mg/L tryptophan, 30 mg/L tyrosine, 10 mg/L uracil) unless otherwise indicated. The following chemicals were obtained from the indicated sources: cycloheximide [Sigma], hippuristanol [a generous gift of Junichi Tanaka, University of the Ryukyus], β-estradiol [Sigma], sordarin [Sigma], 3-indoleacetic acid [Sigma], $IP_6$ [Sigma], 4-thiouracil (4TU) [Arcos], 3-amino-1,2,4-triazole (3AT) [Sigma].

### Plasmid Construction

All plasmids are listed in *Supplementary file 3* and the plasmid sequences are available as *Supplementary file 5*. pNH604-*pGPD1-LexA-EBD-B112* (pKW3908) was constructed by standard restriction enzyme cloning using plasmid FRP880 (*Ottoz et al., 2014*) as a PCR template for LexA-EBD-B112 and pNH603-*pGPD1-LexA-EBD-B112* (pKW4073) was derived from this plasmid. pNH603-*pGDP1-OsTIR1* (pKW2874) and pNH605-*pGPD1-OsTIR1* (pKW2830) were constructed by standard restriction enzyme cloning using pNHK53 as a PCR template for OsTIR1 (*Nishimura et al., 2009*). pFA6a-*IAA7-3V5-KanMx6* (pKW4325) was generated by Gibson assembly using a cDNA pool of Arabidopsis thaliana as template for IAA7. Plasmids pRS425-*p4xLexOcyc1-CDC33*(± ΔG ± ΔCAP)-(±3V5) (pKW4326, pKW4327, pKW4328, pKW4329, pKW4330, pKW4331, pKW4332 and pKW4333) were generated by a combination of Gibson assembly and site-directed mutagenesis using FRP793 (*Ottoz et al., 2014*) as a PCR template for p4xLexOcyc1. Plasmid pRS313-*HR1_Chr2(SCO2)-p4xLex-Ocyc1-3xGST-V5-24xPP7sl-tCYC1-NatNT2-HR2_Chr2(SCO2)* (pKW4910) was constructed using standard restriction enzyme based cloning methods.

### 4TU metabolic labeling and RNA analysis

Cells were grown in CSM-lowURA overnight to post-diauxic stage ($OD_{600}$ = 1–5) and then back-diluted in CSM-lowURA at $OD_{600}$ = 0.1. Cells were grown for at least two doublings, back-diluted to $OD_{600}$ = 0.4 and 1 mM 4TU was added from a 1 M 4TU stock in DMSO. Cells were collected by filtration and immediately snap frozen in liquid nitrogen. Cell pellets were resuspended in 400 μL ice-cold TES buffer (10 mM TrisHCl pH7.5, 10 mM EDTA, 0.5% SDS) containing 5 ng 4TU-*srp1α*(Hs)-polyA spike RNA and 5 ng *rcc1*(Xl)-polyA spike RNA. 400 μL acid-saturated phenol was added and samples were continuously vortexed for 1 hr at 65°C. The aqueous phase was then subjected to an additional phenol extraction followed by chloroform extraction and then isopropanol precipitated. Total RNA was pelleted, resuspended and 14 μg was biotinylated with MTSEA-biotin [Biotium] as previously described (*Duffy et al., 2015*). 10 μg of biotinylated total RNA was then subjected to oligo-dT bead [Life technologies] selection or 500 ng total RNA was used as input for the streptavidin bead selection depending on the experiment. 25 μL MyOne streptavidin C1 Dynabeads [Life technologies] were washed with 25 μL each of 0.1 M NaOH (2x), 0.1 M NaCl (1x) and buffer3 (10 mM TrisHCl pH7.4, 10 mM EDTA, 1 M NaCl) (2x). The beads were then resuspended in 25 μL buffer3 and 2.5 μL 50x Denhardt's reagent was added. Beads were then incubated with gentle agitation for 20 min, washed with 75 μL buffer3 (4x) and resuspended in 25 μL buffer3 with 2 μL 5 M NaCl added. Biotinylated RNAs were added to the blocked streptavidin beads and incubated with

gentle agitation for 15 min. The flowthrough was collected and the beads were washed with 75 µL each buffer3 prewarmed to 65°C (1x), buffer4 (10 mM TrisHCl pH7.4, 1 mM EDTA, 1%SDS) (1x) and 10%buffer3 (2x). The washes were pooled with the flowthrough and 25 µL 5 M NaCl and 15 µg linear acrylamide [Ambion] were added prior to isopropanol precipitation. Biotinylated RNAs were eluted from the streptavidin beads first by a 5 min incubation with gentle agitation in 5% β-mercaptoethanol and a subsequent 10 min incubation at 65°C in 5% β-mercaptoethanol. Eluates were pooled and 7 µL 5 M NaCl and 15 µg linear acrylamide were added prior to isopropanol precipitation. RNAs were DNaseI [NEB] treated prior to downstream analysis. 4TU-*srp1α*(Hs)-polyA and *rcc1*(Xl)-polyA spike RNAs were generated as previously described using plasmids pKW1644 and pKW1643 respectively (*Munchel et al., 2011*).

## RT-qPCR quantification of RNA abundance

2–50 ng of mRNA (depending on sample) was used for reverse transcription using Superscript II [Life technologies] with random hexamer primers. cDNA was quantified on a StepOnePlus Real-Time PCR system using a SYBR green PCR mix [ThermoFisher] with gene specific primers (*Supplementary file 4*).

## High-throughput sequencing and RNAseq quantification

50–100 ng mRNA was used as input material to generate strand-specific sequencing libraries using the NEXTflex Rapid Directional Illumina RNA-Seq Library Prep Kit [BioO] according to the manufacturer's instructions. The libraries were sequenced on an Illumina HiSeq 4000 sequencer by the QB3 Vincent J. Coates Genomics Sequencing Laboratory. Reads were mapped, aligned and quantified using the Tuxedo tools (*Trapnell et al., 2012*).

## Immunoblot analysis

For immunoblot analysis of Cdc33-3V5 and Cdc33-IAA7-3V5, cell were pelleted and resuspended in 5% ice-cold trichloroacetic acid for a minimum of 10 min. The acid was washed away with acetone and cell pellets were air-dried overnight. The pellet was resuspended in 100 µL lysis buffer (50 mM Tris-HCl pH 7.5, 1 mM EDTA, 2.75 mM DTT) and pulverized with glass beads in a beadmill [BioSpec] for 5 min. Sample buffer was added and the lysates were boiled for 5 min. V5 tagged proteins were detected with a mouse anti-V5 antibody [Life technologies, RRID: AB_2556564] at a 1:2000 dilution. Hexokinase (Hxk1) was detected using a rabbit anti-hexokinase antibody [H2035, Stratech, RRID: AB_2629457] at a 1:10,000 dilution. IRdye 680RD goat-anti-rabbit [LI-COR Biosciences, RRID: AB_10956166] and IRdye 800 donkey-anti-mouse [LI-COR Biosciences, RRID: AB_621847] were used as secondary antibodies.

## Data processing and half-life determination

RNA measurements were first normalized by the abundance of the spike RNA (*rcc1*(Xl) for decay measurements and *srp1α*(Hs) for synthesis measurements) and then normalized to the t = 0 value. These data were then fitted to the following decay model:

$$RNA(t) = eff * 2^{-t/T_h} + (1 - eff)$$

Where *eff* is a bulk efficiency term that accounts for the efficiency of 4TU labeling, biotin conjugation and separation of biotinylated RNAs from unbiotinylated RNAs and $T_h$ is the half-life. A script was written in R to perform filtering, normalization and fitting of the decay data (https://github.com/LeonYenLeeChan/RNAdecay; copy archived at https://github.com/elifesciences-publications/RNA-decay). For synthesis rate determination, data were fit to:

$$RNA(t) = k_s * t + offset$$

Where $k_s$ is the synthesis rate and the *offset* is the y-intercept. Please see appendix 1 for more details.

## Growth determination

Cells were grown to mid-log phase, back diluted to OD 0.1–0.3 and growth at 30°C was monitored either manually by measuring absorbance at 600 nm or in a Tecan Infinite M1000 plate reader [Tecan].

## Wide-field fluorescence microscopy

Cells for imaging experiments were grown to exponential phase in CSM + 2% dextrose. Cells were then transferred onto Concanavalin A-treated 96-well glass bottom Corning plates (Corning, Corning, NY). For experiments described in *Figure 4A–C* and *Figure 4—figure supplement 1B–C* cells were visualized at room temperature using the DeltaVision Elite Imaging System with softWoRx imaging software (GE Life Sciences, Marlborough, MA). The system was based on an Olympus 1 × 71 inverted microscope (Olympus, Japan), and cells were observed using a UPlanSApo 100 × 1.4 NA oil immersion objective. Single plane images were acquired using a DV Elite CMOS camera. For experiments described in *Figure 4D–F* and *Figure 4—figure supplement 1D*, cells were visualized at room temperature using an inverted epi-fluorescence microscope (Nikon Ti) equipped with a Spectra X LED light source and a Hamamatsu Flash 4.0 sCMOS camera using a 100x Plan-Apo objective NA 1.4 and the NIS Elements software. Quantification of co-localization was performed on all planes of a 3D stack using the Colocalization Threshold tool in FIJI. Image processing for PB counting was performed using Diatrack 3.5 particle tracking software.

## Spinning disk confocal microscopy

Samples were grown in CMS + 2% dextrose to exponential phase and imaged using an Andor/Nikon Yokogawa spinning disk confocal microscope (Belfast, United Kingdom) with Metamorph Microscopy Automation and Image Analysis software (Molecular Devices, Sunnyvale, CA). The system was based on a NikonTE2000 inverted microscope and cells were observed using a PlanApo100 ×1.4 NA oil immersion objective and single plane images were captured using a Clara Interline CCD camera (Andor).

## Acknowledgements

We are grateful to Dr. Junichi Tanaka for the generous gift of hippuristanol and to Dr. Jörg Stelling for generously sharing plasmids. We would like to thank Joshua Bloom for helpful discussions and advice regarding statistical frameworks. We also thank Dr. Kathy Collins, members of the Ünal, Brar and Weis labs for their critical reading of this manuscript and helpful discussions, and the Brar and Ünal labs for their generous sharing of equipment and reagents. LYC is an HHMI Fellow of the Damon Runyon Cancer Research Foundation and is further supported by a grant from the Shurl and Kay Curci foundation. SH acknowledges support from an EMBO long-term fellowship (ALTF 290–2014, EMBOCOFUND2012, GA-2012–600394 to SH). This work was supported by NIH/NIGMS (R01GM058065 and R01GM101257 to KW) and the Swiss National Science Foundation (SNF 159731 to KW). This work used the Vincent J Coates Genomics Sequencing Laboratory at UC Berkeley, supported by NIH S10 OD018174 Instrumentation Grant.

## Additional information

### Competing interests

Karsten Weis: Reviewing editor, *eLife*. The other authors declare that no competing interests exist.

### Funding

| Funder | Author |
| --- | --- |
| National Institutes of Health | Leon Y Chan<br>Christopher F Mugler<br>Karsten Weis |
| Damon Runyon Cancer Research Foundation | Leon Y Chan |

| Shurl and Kay Curci Foundation | Leon Y Chan |
| European Molecular Biology Organization | Stephanie Heinrich |
| Schweizerischer Nationalfonds zur Förderung der Wissenschaftlichen Forschung | Stephanie Heinrich Pascal Vallotton Karsten Weis |

The funders had no role in study design, data collection and interpretation, or the decision to submit the work for publication.

### Author contributions

Leon Y Chan, Conceptualization, Resources, Data curation, Software, Formal analysis, Funding acquisition, Validation, Investigation, Visualization, Methodology, Writing—original draft, Project administration, Writing—review and editing; Christopher F Mugler, Conceptualization, Data curation, Formal analysis, Investigation; Stephanie Heinrich, Conceptualization, Data curation, Formal analysis, Investigation, Visualization, Methodology; Pascal Vallotton, Software, Formal analysis; Karsten Weis, Conceptualization, Resources, Supervision, Funding acquisition, Investigation, Project administration, Writing—review and editing

### Author ORCIDs

Leon Y Chan http://orcid.org/0000-0002-0189-4689
Christopher F Mugler https://orcid.org/0000-0001-8258-1192
Stephanie Heinrich http://orcid.org/0000-0003-1607-4525
Karsten Weis http://orcid.org/0000-0001-7224-925X

### Decision letter and Author response

Decision letter https://doi.org/10.7554/eLife.32536.078
Author response https://doi.org/10.7554/eLife.32536.079

# Additional files

### Supplementary files

• Supplementary file 1. Sources of data for *Figure 2*.
DOI: https://doi.org/10.7554/eLife.32536.022

• Supplementary file 2. Yeast strains.
DOI: https://doi.org/10.7554/eLife.32536.023

• Supplementary file 3. Plasmids.
DOI: https://doi.org/10.7554/eLife.32536.024

• Supplementary file 4. qPCR primers.
DOI: https://doi.org/10.7554/eLife.32536.025

• Supplementary file 5. Plasmid sequences.
DOI: https://doi.org/10.7554/eLife.32536.026

• Transparent reporting form
DOI: https://doi.org/10.7554/eLife.32536.027

### Data availability

Sequencing data have been deposited in GEO under accession code GSE119560.

The following dataset was generated:

| Author(s) | Year | Dataset title | Dataset URL | Database and Identifier |
|---|---|---|---|---|
| Chan L | 2018 | mRNA stability as measured by thiouracil incorporation in the presence and absence of translational inhibitors | https://www.ncbi.nlm.nih.gov/geo/query/acc.cgi?acc=GSE119560 | NCBI Gene Expression Omnibus, GSE119560 |

The following previously published datasets were used:

| Author(s) | Year | Dataset title | Dataset URL | Database and Identifier |
|---|---|---|---|---|
| Nagalakshmi U, Wang Z, Waern K, Shou C, Raha D | 2008 | mRNA abundance and transcript boundaries | https://www.ncbi.nlm.nih.gov/pmc/articles/PMC2951732/bin/NIHMS229938-supplement-supp__tables.xls | PubMed Central, NIHMS229938-supplement-supp__tables.xls |
| Subtelny AO, Eichhorn SW | 2014 | polyA tail length | https://www.ncbi.nlm.nih.gov/geo/query/acc.cgi?acc=GSE52809 | NCBI Gene Expression Omnibus, GSE52809 |
| Brar GA, Yassour M, Friedman N, Regev A, Ingolia NT, Weissman JS | 2012 | translational efficiency | https://www.ncbi.nlm.nih.gov/geo/query/acc.cgi?acc=GSE34082 | NCBI Gene Expression Omnibus, GSE34082 |
| Drummond DA, Raval A, Wilke CO | 2006 | CAI | https://www.ncbi.nlm.nih.gov/pubmed/16237209 | Molecular Biology and Evolution, en-cai-fop.txt |
| Pechmann S, Frydman J | 2013 | nTE | https://web.stanford.edu/group/frydman/codons/normalizedTE.txt | Frydman Lab, codons/normalizedTE.txt |
| Neymotin B, Athanasiadou R, Gresham D | 2014 | Gresham mRNA halflife, Supplemental table 5 | https://www.ncbi.nlm.nih.gov/pmc/articles/PMC4174445/bin/supp_20_10_1645__index.html | PubMed Central, supp_20_10_1645__index.html |
| Miller C, Schwalb B, Maier K, Schulz D | 2011 | Cramer (1) mRNA halflife | https://www.ncbi.nlm.nih.gov/pmc/articles/PMC3049410/bin/msb2010112-s1.txt | PubMed Central, msb2010112-s1.txt |
| Sun M, Schwalb B, Schulz D, Pirkl N, Etzold S, Larivière L, Maier KC, Seizl M, Tresch A, Cramer P | 2012 | Cramer (2) mRNA halflife | https://www.ebi.ac.uk/arrayexpress/experiments/E-MTAB-760/ | Electron Microscopy Data Bank, E-MTAB-760 |
| Wang Y, Liu CL, Storey JD, Tibshirani RJ, Herschlag D, Brown PO | 2002 | Brown (1) and (2) mRNA halflife | http://www-genome.stanford.edu/turnover/Yuleidatafiles/halflifeTable.txt | Stanford Genomic Resources, Yuleidatafiles/halflifeTable.txt |
| Duttagupta R, Tian B, Wilusz CJ, Khounh DT, Soteropoulos P, Ouyang M, Dougherty JP, Peltz SW | 2005 | Peltz mRNA halflife, Supplementary Tables S1, S2, and S3 | https://www.ncbi.nlm.nih.gov/pmc/articles/PMC1156976/bin/molcellb_25_13_5499__index.html | PubMed Central, molcellb_25_13_5499__index.html |
| Presnyak V, Alhusaini N, Chen YH, Martin S, Morris N, Kline N, Olson S, Weinberg D, Baker KE, Graveley BR, Coller J | 2015 | Coller (1) and (2) mRNA halflife, Table S1 | https://www.ncbi.nlm.nih.gov/pmc/articles/PMC4359748/bin/NIHMS665436-supplement-5.xls | PubMed Central, NIHMS665436-supplement-5.xls |
| Grigull J, Mnaimneh S, Pootoolal J, Robinson MD, Hughes TR | 2004 | Hughes mRNA stability | http://hugheslab.ccbr.utoronto.ca/supplementary-data/Grigull/Fig2A_data.xls | Hughes Lab, University of Toronto, Grigull/Fig2A_data.xls |
| Holstege FC, Jennings EG, Wyrick JJ, Lee TI, Hengartner CJ, Green MR, Golub TR, Lander ES, Young RA | 1998 | Young mRNA stability | http://www.wi.mit.edu/young/expression.html | Young Lab, Whitehead Institute for Biomedical Research, young/expression.html |
| Geisberg JV | 2014 | Struhl mRNA stability, Table S2 | https://www.ncbi.nlm.nih.gov/pmc/articles/ | PubMed Central, NIHMS552811- |

| | | | | PMC3939777/bin/ NIHMS552811-supple-ment-2.xlsx | supplement-2.xlsx |
|---|---|---|---|---|---|
| Shalem O, Dahan O, Levo M, Marti-nez MR, Furman I, Segal E, Pilpel Y | 2008 | Pipel mRNA stability, Supplementary Table 1 | | https://www.ncbi.nlm. nih.gov/pmc/articles/ PMC2583085/bin/ msb200859-s2.xls | PubMed Central, msb200859-s2.xls |
| Munchel SE, Shult-zaberger RK, Taki-zawa N | 2011 | Weis (1) mRNA stability, Supplemental Data | | https://www.molbiolcell. org/doi/suppl/10.1091/ mbc.e11-01-0028/suppl_ file/mc-e11-01-0028-s10. xls | Molecular Biology of the Cell, mc-e11-01-0028-s10.xls |
| Pelechano V, Pérez-Ortín JE | 2010 | Perez-Ortin mRNA stability | | https://onlinelibrary.wi-ley.com/action/down-loadSupplement?doi=10. 1002%2Fyea.1768&file= yea_1768_supportin-forTS1.xls | Wiley Online Library , yea_1768_ supportinforTS1.xls |

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

# Appendix 1

DOI: https://doi.org/10.7554/eLife.32536.028

## Extended technical appendix detailing the development and refinement of the 4TU-chase labeling experiment

To avoid the multiple problems associated with measuring mRNA stability using global transcriptional inhibition, we sought to employ metabolic labeling methods that would allow us to measure these kinetic parameters in a minimally invasive manner. We developed a working method and used this to collect one of the first profiles of mRNA stability in yeast cells (*Munchel et al., 2011*). Briefly, we grew cells to early exponential phase, labeled them with 0.2 mM 4TU for 2 hr and then transferred the cells to fresh media containing 19.6 mM (2.2 mg/mL) uracil. We collected cells from a timecourse during the chase phase and extracted total cellular RNA. We biotinylated these RNAs with HPDP-biotin, selected for the poly-adenylated fraction using oligo-dT beads and finally separated labeled mRNAs (here the 'old-mRNA' pool) from unlabeled mRNAs ('newly-synthesized' pool). Based on follow-up work as well as the lack of agreement between our dataset and other metabolic-labeling based mRNA stability profiles, we sought to reexamine and improve our first-generation method.

To begin to test our method, we made use of the osmotic shock responsive gene, *STL1*. Upon osmotic shock, *STL1* mRNA is rapidly and transiently induced (*Appendix1—figures 1A and B*) (*Alepuz et al., 2003*). We performed our 4TU pulse-chase labeling and then subjected cells to an osmotic shock with the addition of 0.4M NaCl. If the pulse-chase labeling were performing optimally, we would expect that all *STL1* mRNAs would contain only unmodified uracil. Rather, we found that a sizable fraction (about 40%) of *STL1* transcripts were eluted from the streptavidin beads (*Appendix1—figure 2*). This bleed through of a newly-made transcript into the labeled 'old-transcript' pool could be a reflection of a number of possible technical problems. First, this could indicate that the chase phase is inefficient in preventing newly made transcripts from being labeled with 4TU. Second, this could indicate that the conjugation of biotin to the thiolated mRNA could be inefficient. Third, this bleed through could also be a symptom of carryover during the separation of thiolated mRNA from unlabeled mRNA. And fourth, it could be the case that our mathematical model requires modification to account for bleed through which cannot be addressed by experimental optimization.

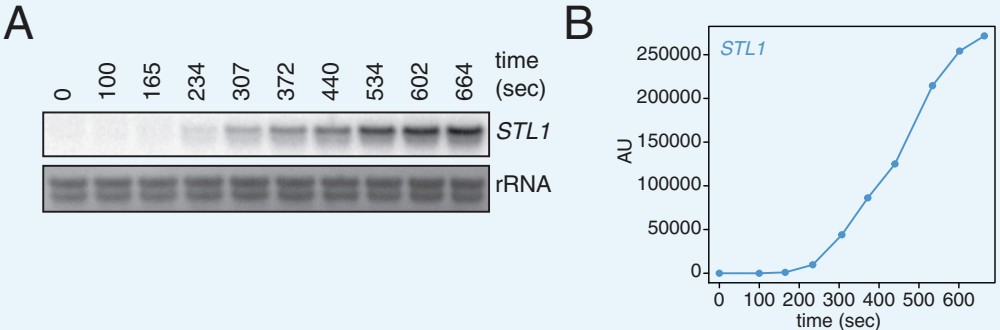

**Appendix 1—figure 1.** Kinetics of *STL1* mRNA induction. (**A**) Wild-type cells (KWY165) were grown to exponential phase, subjected to a 0.4 M NaCl salt shock and samples were collected at the indicated times for northern blot analysis. (**B**) Quantification of data in (**A**). *STL1* mRNA levels were corrected to background and also to rRNA levels.

DOI: https://doi.org/10.7554/eLife.32536.029

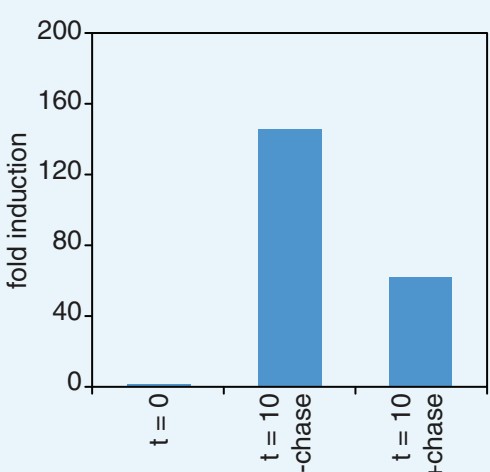

**Appendix 1—figure 2.** Chase inefficiency as revealed by *STL1* induction. Wild-type cells (KWY165) were grown to $OD_{600}$ = 0.2, labeled with 0.2 mM 4TU for 2 hr and then washed out of the labeling media. Half of the cells were returned to the same media and the other half were resuspended in chase media containing 19.6 mM uracil. Both cultures were immediately subjected to 0.4 M NaCl salt shock and samples were collected 10 min after the salt shock. Thio-labeled mRNAs were purified and abundance of *STL1* thio-mRNA was determined by qPCR, normalized to a thio-labeled spike (*srp1α (Hs)*) and then normalized to the levels of *STL1* in pre-salt shocked cells (t = 0).

DOI: https://doi.org/10.7554/eLife.32536.030

The non-linear nature of detection based on affinity capture is inherent to the 4TU method and is a possible contributor to the bleed through problem. Put another way, in theory, an mRNA containing one thio-uracil could be treated the same as an mRNA containing 100 thio-uracils in the context of affinity purification. This is not an issue if the pulse-chase is performing optimally but even a small inefficiency in the chase, specifically new transcripts continuing to incorporate a low level of 4TU, would result in bleed through. We tested this possibility by again making use of the *STL1* system and varied the time of salt-shock and observed the degree to which newly made *STL1* mRNA partitioned between the bead-captured eluate and flowthrough fractions. We found that immediately upon uracil chase, salt shock led to a majority of newly made *STL1* transcripts being retained on the streptavidin beads. As we increased the lag time between uracil chase and salt-shock, we found that an increasing fraction of *STL1* mRNAs partitioned into the flowthrough (***Appendix1—figures 3A and B***). The trend of increasing chase efficiency as time after chase was increased gave us a clear indication that inefficient chase was indeed a problem. We sought to correct this problem by switching from the existing 4TU-pulse/uracil-chase to a 4TU-chase labeling scheme where the non-linear nature of affinity capture-based detection is an advantage rather than a disadvantage. We grew cells to mid-exponential phase ($OD_{600}$ = 0.4) and then added 0.2 mM 4TU to the culture. We again varied the time of salt-shock post 4TU chase and observed how the newly synthesized *STL1* transcript partitioned in the bead capture. We observe that even with no lag phase between 4TU chase and salt-shock, most of the newly synthesized *STL1* mRNA was effectively captured by the streptavidin beads. As the lag time was increased, a greater fraction of newly made *STL1* transcript was captured by the streptavidin beads (***Appendix1—figures 3C and D***).

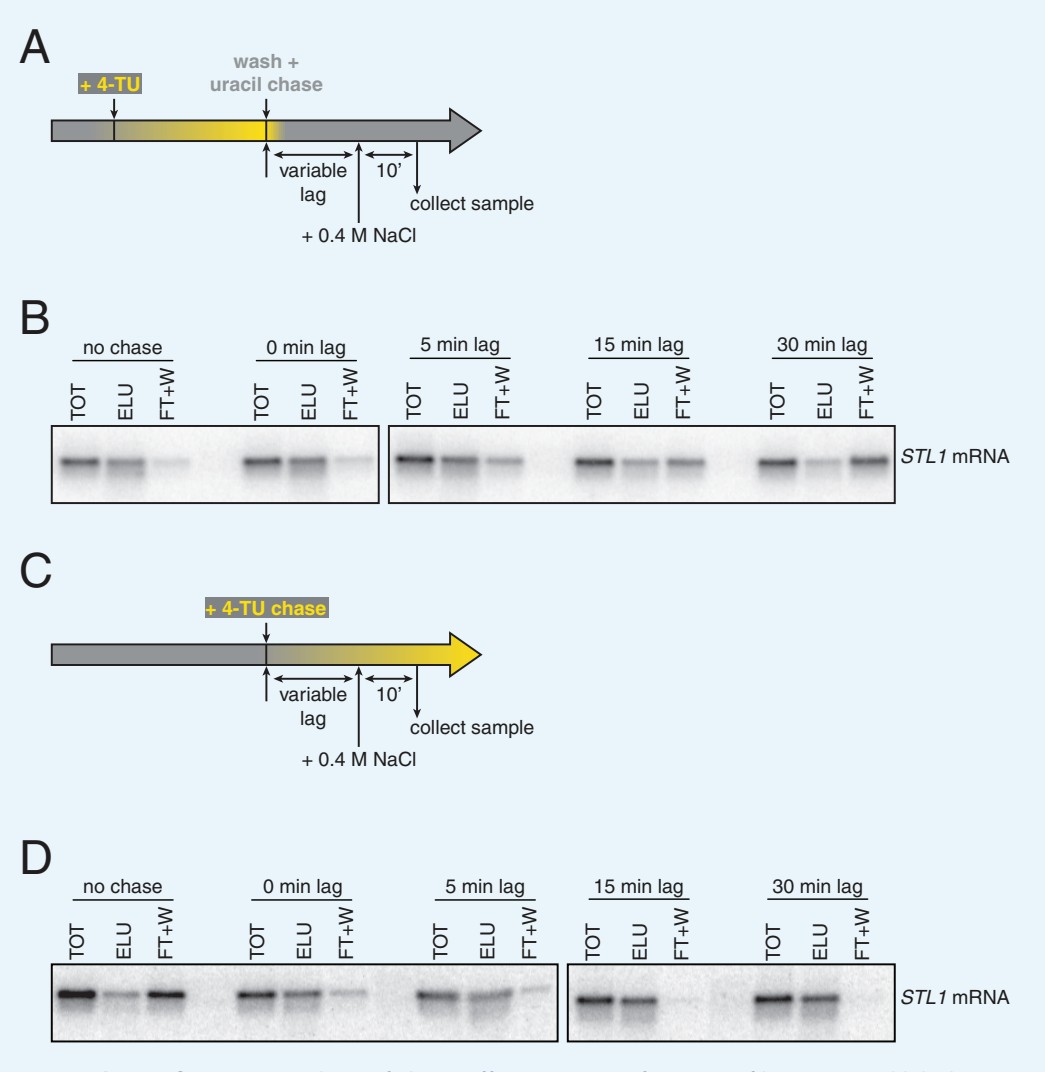

**Appendix 1—figure 3.** Analysis of chase efficiencies as a function of lag time and labeling scheme. (**A**) 4TU-pulse/uracil-chase labeling scheme and experimental setup for (**B**). (**B**) Wild-type cells (KWY165) were grown to exponential phase, the experiment outlined in (**A**) was performed and levels of *STL1* mRNA were determined by northern blot. (**C**) 4TU-chase labeling scheme and experimental setup for (**D**). (**D**) Wild-type cells (KWY165) were grown to exponential phase, the experiment outlined in (**C**) was performed and levels of *STL1* mRNA were determined by northern blot.

DOI: https://doi.org/10.7554/eLife.32536.031

In addition to the switch in labeling scheme, we also tested the possibility that higher concentrations of 4TU in the chase could lead to improved new-transcript labeling. We titrated 4TU concentrations and found that the maximum tolerated dose of 4TU was 1 mM in our CSM-lowURA media (*Appendix1—figure 4*). Higher concentrations inhibited cell growth and would be self-defeating with respect to measuring mRNA dynamics in a minimally-perturbed system. We measured the stability of the *ACT1* transcript using a 0.2 mM 4TU-chase or a 1 mM 4TU-chase. Use of the higher 4TU concentration enabled a more efficient chase as indicated by the lower levels of *ACT1* transcript in the flowthrough fractions especially later in the timecourse (*Appendix1—figure 5*).

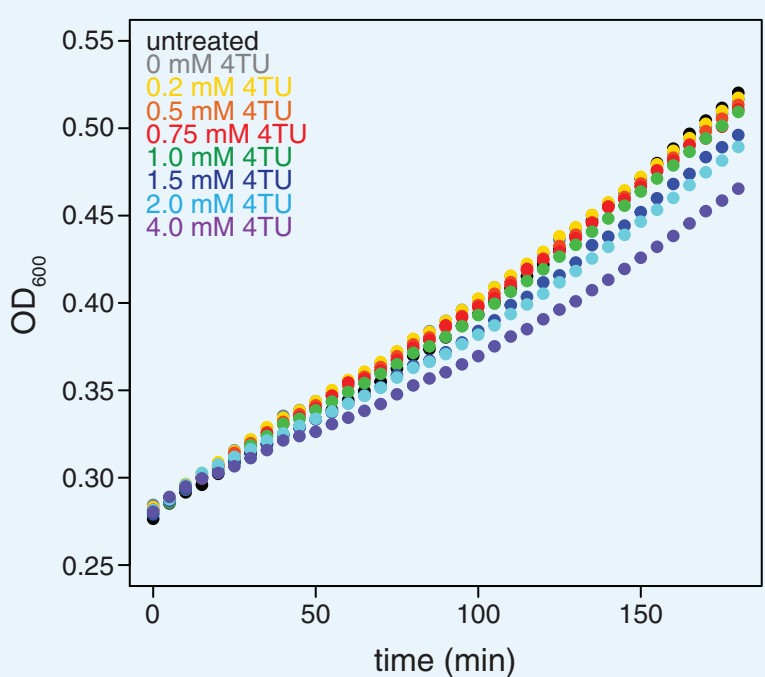

**Appendix 1—figure 4.** Effects of 4TU on cell growth. Wild-type cells (KWY165) were grown to exponential phase, treated with the indicated concentration of 4TU and growth was monitored by absorbance at 600 nm.

DOI: https://doi.org/10.7554/eLife.32536.032

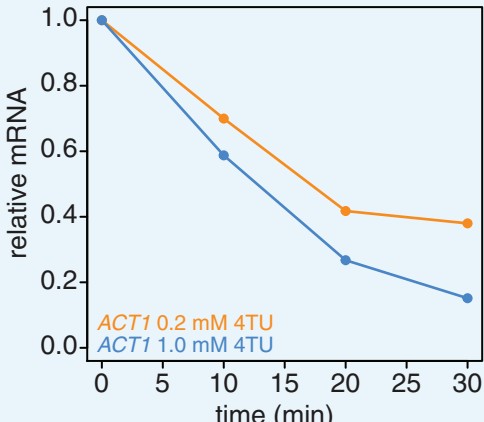

**Appendix 1—figure 5.** Increasing the 4TU concentration during the chase improves the subtraction of newly synthesized mRNAs. Wild-type cells (KWY165) were subjected to the 4TU-chase protocol with either 0.2 mM 4TU in the chase (orange) or 1 mM 4TU (blue) and the unlabeled *ACT1* mRNA levels were quantified.

DOI: https://doi.org/10.7554/eLife.32536.033

We also examined if introducing a lag period between 4TU-chase and beginning the collection of the mRNA stability timecourse improved the efficiency of new transcript labeling. We measured the stability of the *ACT1* mRNA with a range of lag times. We found that even a brief lag period (2 min) improved the efficiency and consistency of the method (*Appendix1—figure 6*). Increasing lag times beyond 2 min did not improve the quality of the data. Moreover, increasing lag times is a balancing act; some lag is beneficial for robust labeling but an overly long lag period fails to capture the dynamics of the most rapidly decayed mRNAs. Thus we opted for the 2 min lag time where we still observed a beneficial effect.

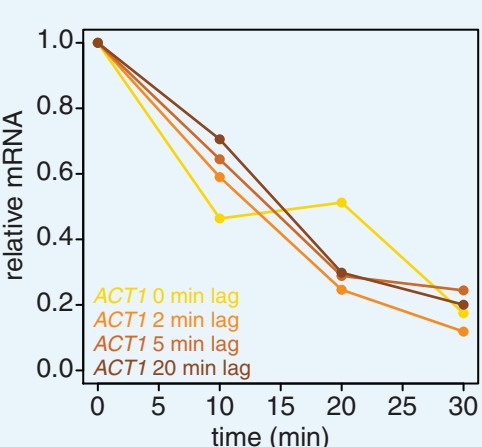

**Appendix 1—figure 6.** Introducing a brief lag period between the 4TU-chase and starting the decay timecourse improves the quality of the decay data. Wild-type cells (KWY165) were subjected to the 4TU-chase protocol. A variable amount of time was allowed to elapse between the 4TU-chase and collection of the t = 0 sample and the unlabeled *ACT1* mRNA levels were quantified.

DOI: https://doi.org/10.7554/eLife.32536.034

Having optimized the labeling conditions, we next turned our attention to potential issues with carryover during the streptavidin bead-based RNA separation. We examined how unlabeled transcripts partitioned between the bead-bound and flowthrough fractions. We found that indeed unlabeled RNAs were being retained on the beads and moreover, different RNAs were retained to different degrees (compare *Appendix1-figure 3D* lanes 1–3 for *STL1* and *Appendix 1—figure 7* lanes 1–3 for *rcc1 (Xl)*). This is especially problematic as it implies that in addition to a global distortion of mRNA stability, bead carryover can produce transcript specific artifacts as well thus complicating even relative comparisons of mRNA half-lives. We reasoned that improvements to both bead blocking as well as bead washing could mitigate this carryover problem. We tested a variety of blocking agents such as single-stranded salmon sperm DNA, polyA RNA, heparin and Denhardt's reagent in addition to bacterial tRNAs as was used in our first-generation method. We found that bead-blocking with Denhardt's reagent resulted in a marked improvement in unlabeled RNA carryover. Moreover, we found that this bead-blocking method did not interfere with the capture of an in vitro generated thio-labeled *srp1α (Hs)* RNA (*Appendix1-figure 7*). We also examined how unlabeled RNAs were washed off the beads in the wash steps following streptavidin bead capture. We found that most of the unlabeled RNA was contained in the flowthrough fraction. A fraction of remaining RNA that stuck to the beads was released with a high-salt 65C wash step but further washes with this buffer were ineffective in releasing additional non-specifically bound RNAs. We took advantage of the fact that the biotin-streptavidin interaction is resistant to SDS concentrations as high as 3%. With the addition of a single 1% SDS wash, we were able to release the majority of the remaining unlabeled RNA from the beads (*Appendix1-figure 8*). Additional washes with this buffer did not release a significant amount of unlabeled RNAs. Again, we found that this more stringent wash protocol did not interfere with the ability of the streptavidin beads to properly capture labeled RNAs.

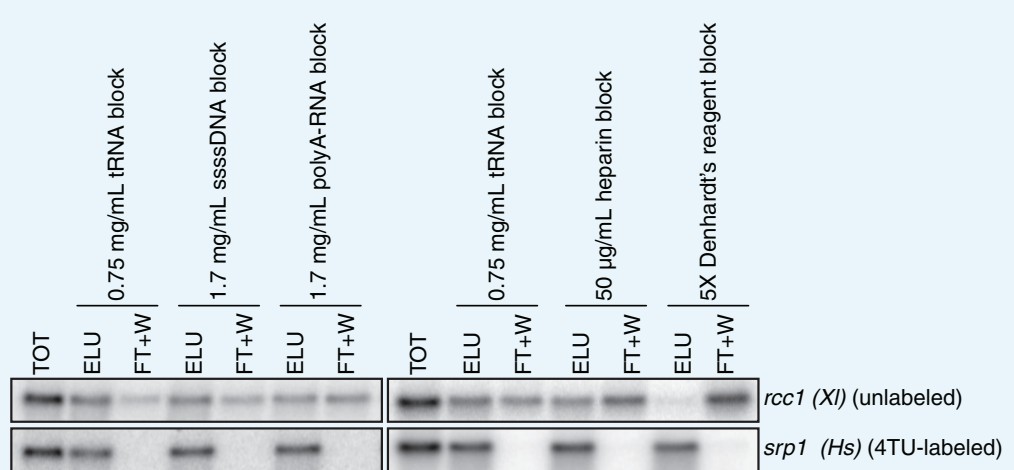

**Appendix 1—figure 7.** Analysis of streptavidin bead blocking efficiency. Wild-type cells (KWY165) were labeled with 1 mM 4TU for 2 hr. 5 ng of in vitro transcribed *rcc1 (Xl)* mRNA spike and 5 ng of in vitro transcribed 4TU-labeled *srp1α (Hs)* mRNA spike were added to 10 µg of extracted total RNA and the RNA mix was biotinylated. mRNAs were enriched for using oligo-dT beads and the mRNAs were then subjected to streptavidin bead selection. Streptavidin beads were prepared using the indicated blocking agents and the total (TOT), eluate (ELU) and flowthrough plus washes (FT + W) fractions were analyzed by northern blot.
DOI: https://doi.org/10.7554/eLife.32536.035

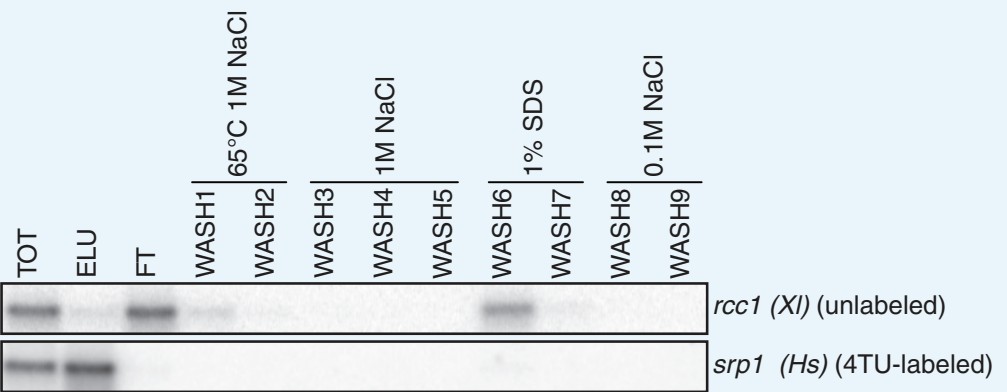

**Appendix 1—figure 8.** Analysis of unlabeled mRNA release during streptavidin bead purification. RNA mixtures were prepared as in *Appendix1—figure 7* and total (TOT), eluate (ELU) flowthrough (FT) and wash (WASH1-WASH9) samples were analyzed by northern blot.
DOI: https://doi.org/10.7554/eLife.32536.036

Recently, an improved biotin crosslinker, MTSEA-biotin, was developed and we tested if MTSEA-biotin performed better in our second-generation metabolic labeling method compared to the standard HPDP-biotin (*Duffy et al., 2015*). We collected a decay timecourse and processed the RNA in parallel only differing the biotin-crosslinker and analyzed the resulting decay curves. We found that the MTSEA-biotin resulted in better labeling evidenced by the reduced unlabeled mRNA levels we observed later in the timecourse (*Appendix1—figures 9A and B*).

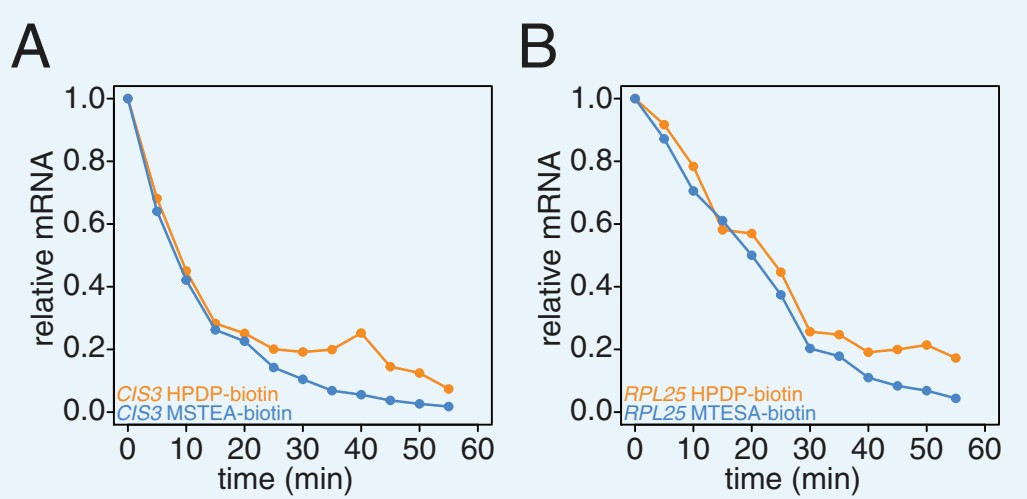

**Appendix 1—figure 9.** Comparison of HPDP-biotin with MTSEA-biotin in the efficiency of subtracting newly synthesized mRNAs during the chase phase. Wild-type cells (KWY165) were subjected to the 4TU-chase protocol and RNAs were biotinylated with either HPDP-biotin (orange) or MTSEA-biotin (blue). Levels of unlabeled *CIS3* and *RPL25* mRNAs were determined.

DOI: https://doi.org/10.7554/eLife.32536.037

Lastly, we revisited our computational model and examined the effects of what remaining inefficiencies in labeling and capture might have on simulated decay curves. We observed that as the efficiency of labeling and capture decreases, the curves become shallower and plateau at a non-zero steady state value (*Appendix1—figure 10A and C*). Note that for a long lived transcript, fitting these data to a single exponential decay model can result in dramatically longer half-lives with little loss of goodness of fit (*Appendix1—figure 10B*). In the case of very unstable transcripts, the single exponential decay model completely fails for even modest levels of inefficiency (*Appendix1—figure 10D*). To properly model these data, we introduced an efficiency of labeling and capture parameter to the standard decay model:

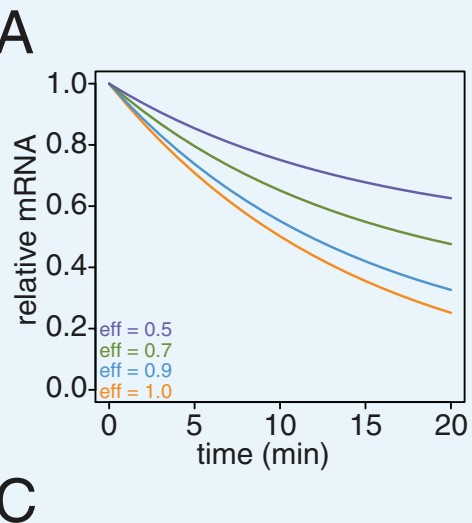

**B**

computed half-life using a
single exponential decay model

| eff | half-life (min) | R² |
|-----|-----------------|------|
| 1.0 | 10 | 1.000 |
| 0.9 | 12.3 | 0.999 |
| 0.7 | 18.7 | 0.993 |
| 0.5 | 29.7 | 0.986 |

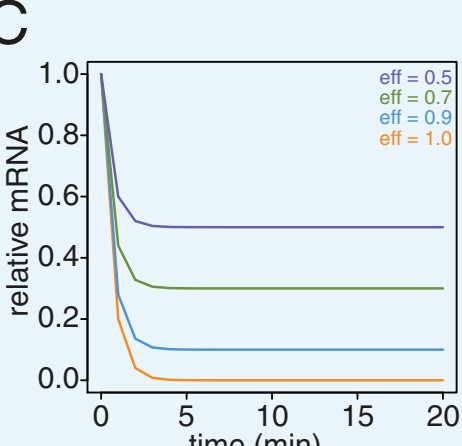

**D**

computed half-life using a
single exponential decay model

| eff | half-life (min) | R² |
|-----|-----------------|------|
| 1.0 | 0.4 | 1.000 |
| 0.9 | 15.1 | 0.280 |
| 0.7 | 32.6 | 0.237 |
| 0.5 | 59.6 | 0.219 |

**Appendix 1—figure 10.** Simulation of the effects of non-ideal efficiencies in labeling and strep-tavidin bead separation on decay kinetics. (**A**) Decay data for an mRNA with a 10 min half-life were simulated with variable degrees of labeling and purification efficiencies. (**B**) The simulated data in (**A**) were fit to a single exponential decay model (RNA(t)=RNA(0)*2$^{t/hl}$ where hl is the half-life) and the half-lives and goodness of fits (R²) were determined. (**C**) Decay data for an mRNA with a 0.4 min half-life were simulated with variable degrees of labeling and purification efficiencies. (**D**) The simulated data in (**C**) were fit to a single exponential decay model (RNA(t)=RNA(0)*2$^{t/hl}$ where hl is the half-life) and the half-lives and goodness of fits (R²) were determined.

DOI: https://doi.org/10.7554/eLife.32536.038

$$RNA(t) = eff * 2^{-t/T_h} + (1 - eff)$$

where $T_h$ is the half-life and *eff* is a bulk efficiency of labeling and thio-RNA capture. Note that in the limiting case where the efficiency approaches 1, the equation reduces to the single exponential decay model. We used this two parameter model for all of our half-life determinations. Moreover, we did not use a global efficiency parameter but rather fit each transcript to its own efficiency as there are transcript specific contributions to this bulk parameter such as bead-stickiness and uracil content.

We made one final modification to the 4TU method with regard to spike-in control RNAs that normalize for differences in the various RNA recovery and bead binding steps of the 4TU protocol. In our first generation protocol, we spiked in control RNAs relative to a constant amount of total RNA which is comprised mostly of rRNA. We then accounted for the mRNA 'decay by dilution' due to cell growth and division by correcting for the growth rate. This method introduces the errors of growth rate determination as well as RNA quantification into

the protocol. We reasoned that in theory, adding our spike-in controls relative to a constant **culture volume**, would eliminate the need to correct for 'decay by dilution.' The only caveat to this approach is the assumption that the efficiency of cell lysis is relatively constant. We set out to test this assumption. We found that for lysing six identical cell pellets that there was a 3% standard deviation in RNA recovery (*Appendix1—figure 11*). We also tested the linearity of recovery of RNA from increasing sizes of cell pellets and found that that the recovery of RNA was robustly linear to input cell material over the range of our experiments (*Appendix1—figure 12*). We concluded that spiking relative to culture volume is an improvement on the first-generation metabolic labeling protocol.

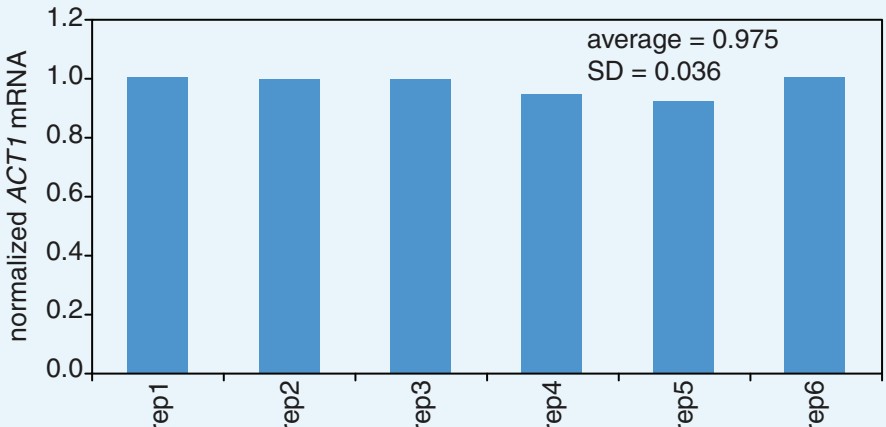

**Appendix 1—figure 11.** Reproducibility of cell lysis during RNA extraction. Wild-type cells (KWY165) were grown to exponential phase and six technical replicate cell pellets of 5 $OD_{600}$ units were collected. 10 ng of in vitro transcribed *rcc1 (Xl)* mRNA spike was added to each sample and the cells were subjected to the RNA extraction protocol. Levels of *ACT1* and the spike mRNAs were determined and the ratio of *ACT1*:spike normalized to replicate one is plotted.

DOI: https://doi.org/10.7554/eLife.32536.039

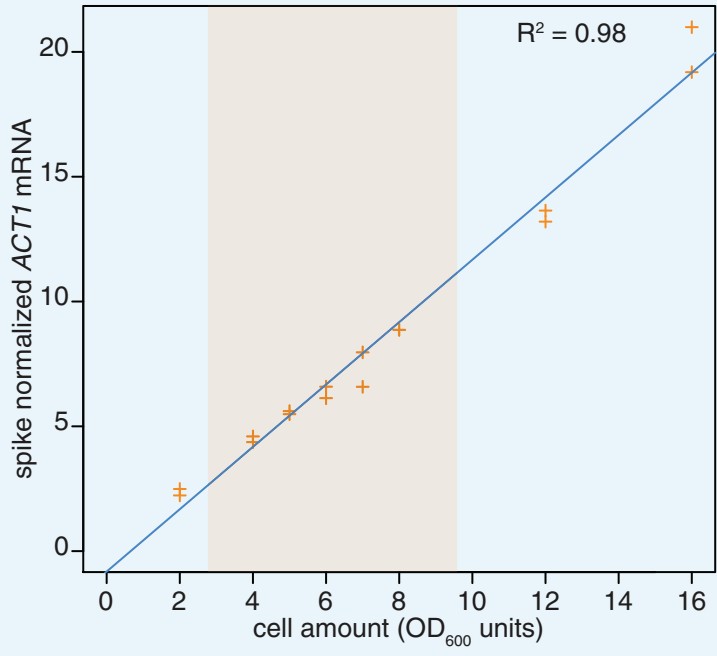

**Appendix 1—figure 12.** Reproducibility of cell lysis during RNA extraction. Wild-type cells

(KWY165) were grown to exponential phase and two technical replicate cell pellets of varying $OD_{600}$ units were collected. 10 ng of in vitro transcribed *rcc1 (Xl)* mRNA spike was added to each sample and the cells were subjected to the RNA extraction protocol. Levels of *ACT1* and the spike mRNAs were determined and the ratio of *ACT1*:spike is plotted. The shaded area indicates the $OD_{600}$ range in which the 4TU-chase experiments are performed.
DOI: https://doi.org/10.7554/eLife.32536.040

In total, we re-examined each step of our 4TU protocol and made improvements where possible. We then introduced a modified mathematical model to account for the remaining inefficiencies in the protocol that we could not experimentally correct. This has resulted in an improved second generation 4TU protocol that is accurate and reproducible.

## Experimental Procedures

### Northern blot analysis

Northern blot experiments were performed as previously described (*Carroll et al., 2011*). Oligonucleotide probes are as follows:

*STL1* GATTTTGGGACCTGCCTCTGGAGAACAAACTTGACAGTG
*rcc1 (Xl)* GAAAGACCAAAGCCATATACATGGCCTTCTTGGGACACCGC
*srp1α (Hs)* GGGCTGTTTTTCTCTGGAAAGTAGTTTCCTGGCAGCTTGAG

