## [Decision Letter]

Thank you for submitting your article "Non-invasive measurement of mRNA decay reveals translation initiation as the major determinant of mRNA stability" for consideration by *eLife*. Your article has been reviewed by James Manley (Senior Editor), a Reviewing Editor, Alan Hinnebusch, and three reviewers three peer reviewers, The following individual involved in review of your submission has agreed to reveal his identity: Roy Parker (Reviewer #3).

The reviewers have discussed the reviews with one another and the Reviewing Editor has drafted this decision to help you prepare a revised submission. The essential revisions may take more than the usual two months we suggest for return of a revised manuscript but given the enthusiasm for this work, we are willing to extend that limit if it proves necessary. It would help to have your response to the essential revisions and an estimate of the time it will take to complete these tasks.

Summary:

Chan et al. present an improved method for measuring the kinetics of mRNA synthesis and degradation that minimizes perturbation of cellular state and that accounts for various potential sources of non-biological variation. They apply this method to measure the half-lives of each type of mRNA in yeast and produce results that are strikingly different from those produced by earlier approaches. In particular, they find that average half-lives are substantially shorter than previously measured. Seeking to explain gene-by-gene variation in mRNA stability, the authors identify rates of translation initiation and optimal codon content as having similarly strong positive correlations with half-life. To distinguish between the more historical model in which the presence of ribosomes protects messages from decay machinery and a newer model in which slow elongation (e.g. of non-optimal codons) serves as a general trigger of decay, the authors measure mRNA stability after applying a variety of chemical and genetic methods of interfering with elongation or with initiation. They find that slow elongation stabilizes mRNA, while slow initiation destabilizes mRNA, favoring a model in which competition for access to the cap, rather than sensing of slow moving ribosomes, serves as the primary gatekeeper for decay pathways.

Essential revisions:

The most important issue to be addressed is whether your use of poly(A) selection has led you to measure the half-lives of only a fraction of mRNAs with poly(A) tails long enough for oligo-dT selection or, in the limit case, to measure rates of deadenylation only. It is necessary that you alter your technique to avoid poly(A) selection by using ribo0 depletion as an alternative approach and determine whether your major conclusions about mRNA half-lives and the relatively greater importance of translation initiation versus elongation rates in determining half-lives still hold.

Second, the reviewers were not convinced of the statistical significance of differences in half-lives of specific mRNAs being claimed in the Results, requiring more thorough documentation of replicates with the appropriate statistical analyses.

Third, because your most significant results at odds with the previous model positing that mRNA decay is triggered by stalled ribosomes containing an empty A-site came from analyzing turnover rates in gcn2 cells treated with 3AT, this approach should be conducted genome-wide to analyze the correlation (or lack thereof) between ribosome stalling at His codons and mRNA half-life.

Fourth, for the experiments using hippuristanol or dominant-negative eIF4E/degron depletion of eIF4E to inhibit translation initiation, it should be verified that the translation initiation rate was reduced for the specific mRNAs whose half-lives are being measured.

Fifth, additional revisions of the text are required to reconcile your model with the results of previously published experiments on reporter mRNAs that provided strong evidence that codon optimality is a strong determinant of mRNA half-life, which would suggest that this mRNA feature has not merely co-evolved with translation initiation efficiency in the manner you suggest.

Sixth, it seems necessary to address the alternative interpretation of your experiments on P-bodies raised by reviewer #1.

Reviewer #1:

This paper describes an improved, non-invasive method for genome-wide measurements of mRNA half-lives in growing yeast cells. The results suggest that mRNA half-lives are between 3-fold and 26-fold shorter than determined previously by other approaches relying on metabolic labeling or inhibition of transcription, respectively. By examining the effects of inhibiting translation elongation or initiation on mRNA half-lives in various ways, that conclude that the rate of translation initiation is a more important determinant of mRNA half-life than is the rate of elongation, at odds with recent reports from the Coller and Green labs indicating that codon optimality is the principle determinant of mRNA stability of native mRNAs in budding yeast. They provide evidence that stalling ribosomes during elongation with the inhibitors cycloheximide or sordorin, or by starving for the amino acid histidine, tends to decrease rather than increase rates of mRNA turnover, presumably by having stalled ribosomes protect the mRNA from nucleolytic digestion, rather than inducing turnover via Dhh1 recognition of stalled ribosomes as recently concluded by Coller and Green. By contrast, an inhibitor of eIF4A or mutants that reduce eIF4F function and presumably produce a global decrease in the initiation rate lead to increased mRNA turnover. Thus, they propose that the rate of initiation is the key determinant of mRNA stability, with a reduction in initiation either exposing the mRNA cap to the decapping machinery or decreasing ribosome densities on coding regions. Based on the fact that inhibiting eIF4A with hippuristanol (HIP) induces Dhh1-dependent P-bodies, and that the abundance of a stem-loop reporter mRNA in P bodies decayed over time, they propose that inhibiting initiation sends mRNAs to P bodies and that this is, or can be, the site of their decay.

- The differences in half-life claimed for ACT1 and CIS3 mRNAs in response to CHX treatment in Figure 3—figure supplement 1 and Figure 3B are unconvincing, and they have provided no half-life values for these experiments. The claim that RPS2, RPL25, CIS3 and OSW5 mRNAs exhibit slower turnover on 3AT treatment are similarly unconvincing.

- The most significant results presented here that are at odds with the proposal of Coller/Green, that decay is triggered by stalled ribosomes containing an empty A-site, came from analyzing turnover rates in gcn2 cells treated with 3AT to induce histidine starvation and attendant stalling by elongating ribosomes at histidine codons genome-wide. The experiments using CHX or sordorin are not relevant because these drugs do not stall ribosomes with an empty A site. As such, in order to justify their claim that inhibiting elongation in this manner slows down rather than accelerates mRNA decay on a genome-wide level, it is necessary to obtain mRNA turnover data for the entire transcriptome using their technique in 3AT-treated gcn2 cells, rather than examining only a dozen mRNAs by RT-qPCR. In addition, because the previous ribosome profiling experiments using 3AT carried out in the Green lab, which they cite for the effect of 3AT in stalling ribosomes, employed a GCN2+ strain, it is necessary that they confirm that treating a gcn2 mutant with 3AT will cause a similar (or even greater) degree of pauses at histidine codons by elongating ribosomes, and to demonstrate a widespread lack of correlation between pausing at histidine codons measured by ribosome profiling and mRNA half-lives measured by their technique in 3AT-treated cells.

- The authors have used treatment with hippuristanol (HIP), or overexpression of a dominant-negative eIF4E variant lacking cap-binding activity together with degron depletion of eIF4E, as alternative means to inhibit global translation initiation rates. However, they have relied only on cell growth assays to document these effects. It is necessary to show by a more direct assay that translation initiation is impaired for the specific mRNAs whose half-lives are being measured (ACT1, CIS3, and RPL25). The best approach would be to show that the average number of ribosomes per mRNA is reduced on these mRNAs by the treatments shown in Figure 3H-I by probing for these mRNAs across fractionated polysomes; the effects on abundance and average size of bulk polysomes will be obtained in parallel. Alternatively, they could show that the amounts of these mRNAs associated with eIF4G (RNA-IP) are reduced by the two treatments. Treatment with sordarin or CHX would be a key negative control for either approach.

- In Figure 3—figure supplement 2C-D, there are unexplained increases in synthesis of RPL25 in response to CHX and sordorin that require comment.

- The interpretation of the results in Figure 4E, that the reporter mRNA is being degraded in P-bodies, seems to overlook the possibility that the reporter mRNA exchanges rapidly between P-bodies and other compartments, in which case it could be degraded elsewhere, e.g. on polysomes, and its abundance in P-bodies would merely indicate the overall cellular level of the transcript. Is there a way to eliminate this alternative?

- The validity of this method for measuring mRNA half-lives depends on highly efficient incorporation of 4-thioU into all newly synthesized mRNAs, and also highly efficient separation of these mRNAs from the un-labeled mRNAs pre-existing before the 4-thioU pulse, as it is the decay of the later over time that is quantified to determine half-lives. Extracting decay rates from the data requires a theoretical model for the decay, which includes an "efficiency parameter". It is difficult to assess the validity of this model, and hence, the mRNA half-lives determined using it. This is worrisome because their measured half-lives are 3-fold shorter than those measured by other labs using non-invasive metabolic labeling, and there is no attention given to resolving these discrepancies. The authors have provided and extended technical supplement on the refinement of their technique, but it seems important to provide justification for the validity of the modeling and method for extracting turnover times in the main text of the paper.

Reviewer #2:

Chan et al. present an improved method for measuring the kinetics of mRNA synthesis and degradation that minimizes perturbation of cellular state and that accounts for various potential sources of non-biological variation. They apply this method to measure the half-lives of each type of mRNA in yeast and produce results that are strikingly different from those produced by earlier approaches. In particular, they find that average half-lives are substantially shorter than previously measured. Seeking to explain gene-by-gene variation in mRNA stability, the authors identify rates of translation initiation and optimal codon content as having similarly strong positive correlations with half-life. To distinguish between the more historical model in which the presence of ribosomes protects messages from decay machinery and a newer model in which slow elongation (e.g. of non-optimal codons) serves as a general trigger of decay, the authors measure mRNA stability after applying a variety of chemical and genetic methods of interfering with elongation or with initiation. They find that slow elongation stabilizes mRNA, while slow initiation destabilizes mRNA, favoring a model in which competition for access to the cap, rather than sensing of slow moving ribosomes, serves as the primary gatekeeper for decay pathways.

This is important work that has been carefully designed and executed. The unexpectedly strong connection between codon optimality and mRNA decay has emerged as a major area of research over the last three years, and intense efforts are underway to understand the molecular mechanisms that underpin this connection. By calling into question one of the central assumptions of this new sub-field (namely, that slowing down elongation should destabilize mRNA), Chan et al., trigger a messy but necessary rethinking of our understanding of these phenomenon.

1) Presnyak et al., 2015 argues that enriching for mRNA by polyA pulldown produces artificially depressed estimates of half-lives due to the existence of stable deadenylated messages. Why did the authors use polyA pulldown in their protocol? Could this effect contribute to the shorter half-lives they observe? (It is unclear exactly which datasets from this paper are the two Coller datasets in Figure 1—figure supplement 1G.)

2) The authors argue that, given their results, the observed positive correlation between optimal codon content and half life is likely to be due to co-evolution rather than direct causal connection. Several papers (PMIDs: 25768907, 27436874, 27641505, and others), however, have presented compelling evidence for a direct causal connection by showing that systematically deoptimizing the codon content of a reporter construct systematically reduces its stability. To some extent, the authors' results contradict the standard interpretation of these published experiments. The paper would be strengthened if the authors engaged with this contradiction head-on by more thoroughly discussing exactly which ideas in this field are and are not consistent with their new data.

Reviewer #3:

This manuscript addresses the relationship between steps in translation and mRNA degradation in yeast. The manuscript addresses an important and timely issue in the field: Is decay rate coupled to stalled ribosomes, or to poor translation initiation? This has become an issue because of two competing lines of work. First, previous work using reporter mRNAs showed that translation initiation was in competition with decay (both deadenylation and decapping). However, more recent work, using both genome wide analyses and reporter mRNAs has argued that inefficient translation elongation (as defined by non-optimal codons) is the predominant driver of mRNA decay rates. Thus, resolving this issue will be important for understanding mRNA decay control.

The strength of this manuscript is that is uses genome wide analyses of mRNA decay rates both under normal conditions, and with perturbations that affect ribosome elongation rates or translation initiation rates. The main conclusion from the work is that slow ribosome movement actually stabilizes mRNAs, while decreasing translation initiation increases mRNA degradation. This is a useful contribution and worthy of consideration for publication in *eLife*, although there are substantive issues that should be addressed before publication.

1) The biggest issue with the manuscript is that the procedure for mRNA decay rates involves an oligo(dT) selection step. Since many mRNAs in yeast exist at steady state as oligoadenylated or deadenylated mRNAs (see Herrick et al., 1990 MCB), this only selects for a substep of mRNAs (typically with A tails longer than 30). Thus, the approach is really only measuring the deadenylation rate of mRNAs and not overall decay. This is presumably why their decay rates are faster than what has been described previously using methods that do not rely on selection of A+ mRNA species. The best approach for the authors here would be to repeat the experiments using ribo0 RNA, which would allow them to access the full mRNA decay pathway. Alternatively, they could re-write the manuscript as simply studying deadenylation, but this would diminish the impact of the work.

2) In Figure 3B, the differences in decay curves are slight and would need to be supported by error bars to convince the reader these are meaningful. In fact, the data might be more convincing if decay rate was measured with ribo0 RNA since previous work (Beelman and Parker, 1995 and others) had shown cycloheximide primarily inhibits decapping with little effect on deadenylation rate.

[Editors' note: further revisions were requested prior to acceptance, as described below.]

Thank you for resubmitting your work entitled "Non-invasive measurement of mRNA decay reveals translation initiation as the major determinant of mRNA stability" for further consideration at *eLife*. Your revised article has been favorably evaluated by James Manley (Senior Editor), a Reviewing Editor, Alan Hinnebusch, and three reviewers.

All three reviewers agreed that the paper has been significantly improved by the new experiments that were carried out, as well as the revisions of text; and your efforts to address all of the major criticisms of the first version were greatly appreciated. In particular, the new genome-wide data obtained from analyzing total RNA generally supports the conclusions reached from poly(A) selection, which resolves the most important criticism of the original version of the paper.

There are, however, some remaining issues that require your attention, raised by reviewer #1, including the lack of statistical significance of small differences in mRNA half-lives for certain mRNAs, and the apparent lack of any replicates for the new half-life measurements conducted with total RNA. There is also an objection to your interpretation of the new results presented to establish a shift to smaller polysomes of mRNAs in response to down-regulating eIF4E/4G; and a concern that these experiments might have been conducted only once without replication. It is felt that these shortcomings detract from the overall scientific quality of the work and do not satisfy the journal's rigorous standards for replicate measurements and statistical analyses of data. Finally, you have been asked to consider whether the magnitude of the observed effects of reducing initiation or elongation in decreasing or increasing mRNA stabilities, respectively, are substantial enough to account for the order-of-magnitude-range of mRNA stabilities observed in wild-type cells.

Reviewer #1:

It's unfortunate that they chose not to use ribo0 to deplete rRNA, as only a few percent of the total mRNA reads they obtained in the new experiments come from mRNA, and they claim that the relatively weak correlation between mRNA half-lives for total vs polyA selected mRNA is attributable in part to the low read depth for total RNA. Nevertheless, they find the same very short half-life for bulk total mRNA observed for bulk polyA(+) mRNA, which supports the validity of their measurements using polyA selected mRNAs.

There are remaining issues however concerning the changes in half-lives of specific mRNAs in response to different drugs or genetic manipulations. They did no significance testing on these measurements but based on the mean values and standard deviations provided for 3 biological replicates, my own application of a standard t-test indicates that the changes shown in Figure 3B for low-level CHX treatment are not statistically significant (p>0.05) for any of the three mRNAs (ACT1, CIS3, RPL25). This is also the case for PDC1 and CIS3 in response to 3AT in Figures 3D and 3F (one of which was claimed to show a difference, but it wasn't indicated which one), for ACT1 and RPL25 in response to HIP in Figure 3I, and for ACT1 in response to the eIF4E/4G_down mutations in Figure 3J. It could be argued that the new measurements conducted on analysis of total RNA shown in Figure 3—figure supplement 3 can be used to bolster their conclusions, but these measurements were done for only a single biological replicate; moreover, the findings for total RNA for ACT1 again seem to show no difference in half-life for +/- CHX (panel B), and none as well for ACT1 and RPL25 in response to 3AT (panel D). (CIS3 is stabilized in panel D, but it's unclear whether CIS3 was judged to be stabilized in the polyA(+) measurements, as noted above.) Two additional replicates are needed for the total RNA measurements, allowing them to provide mean and SD values. In addition, they need to indicate which differences in Figure 3 and Figure 3—figure supplement.3 (once more replicates are conducted for the latter), have a P-value of <0.05 in the appropriate significance test.

There are also issues with the new experiments added to show that the initiation rates of particular mRNAs are reduced in response to HIP or eIF4E/4G_down. It appears that the experiments were done only once, as there is no mention of data from replicates. Also, the data in panel D of Figure 3—figure supplement.5 for all three mRNAs are quite unexpected, as the mRNA levels are greatly reduced, and the ACT1 and CIS3 mRNAs actually do not shift to smaller polysomes, in response to eIF4E/4G_down. By comparison, the HIP treatment in panel B shows no substantial change in overall mRNA level despite similar decreases in half-life for these mRNAs in the two different treatments and does exhibit the expected shift of all 3 mRNAs to smaller polysomes. It needs to be shown that the results in panels B and D are reproducible in at least one replicate experiment, and the unexpected data for the eIF4E/4G_down experiment need to be discussed.

The median shifts in half-life for all mRNAs shown in Figures 3H and 3K are only 1-2 min on 3-AT or HIP treatment; and with CHX treatment, it appears that the shift is restricted to the mRNAs with the longest half-lives. It seems possible therefore that the effects of reduced initiation or elongation in decreasing or increasing mRNA stability, respectively, are not really strong enough to explain the order of magnitude range of mRNA stabilities observed in wild-type cells. Perhaps the authors should consider softening statements about initiation being the primary determinant of mRNA stability and instead identify it as one important determinant. Is it possible that on some mRNAs, the presence of suboptimal codons is equally important, with ribosomes containing empty A sites being recognized by Dhh1 in the manner suggested by Coller and Green? After all, two out of 12 mRNAs were found to decay faster, not slower, in the presence of 3AT; and presumably there are many other mRNAs behaving like these two mRNAs, in a manner consistent with the Coller/Green Dhh1 model.

- Subsection “An improved non-invasive metabolic labeling protocol reveals that the yeast transcriptome is highly unstable”: Perhaps step 7 is intended?

- Subsection “An improved non-invasive metabolic labeling protocol reveals that the yeast transcriptome is highly unstable”: Indicate the medium used, e.g. nutrient-rich medium with glucose as carbon source?

- Figure 1—figure supplement 1 has 9 panels. Relevant panels should be cited individually in the text-otherwise the reader has to inspect the entire figure without the guidance of text to find the supporting results being cited. This should be fixed throughout the manuscript.

- Subsection “An improved non-invasive metabolic labeling protocol reveals that the yeast transcriptome is highly unstable”: The pool of polyadenylated mRNAs should be designated polyA(+) mRNA rather than polyA-mRNA, which could be interpreted as polyA(-).

- Figure 1—figure supplement 2B: Indicate whether the correlation value of 0.44 is r or r2.

- Subsection “Slowing translation elongation protects transcripts against degradation”: Weinberg and Bartel showed that mRNA abundance is strongly correlated with translation efficiency, so the correlation with abundance could also be reflect the correlation of half-life with TE.

-Was Figure 2—figure supplement 1 cited in Results section?

- Subsection “Slowing translation elongation protects transcripts against degradation” and Figure 3—figure supplement 1A and Figure 3B: It needs to be stated in Results section or at least Figure 3 legends that the data for the specific mRNAs was obtained by RT-qPCR quantification of mRNAs obtained from the mRNA stability profiling technique outlined in Figure 1. There are no error bars for the 50μg/ml CHX experiment in Figure 3—figure supplement 1A, indicating the apparent lack of replicates. As noted above, assuming that the error bars in Figure 3B (which were not defined in the legend) are the SD values for n=3, none of the differences in half-life for the 3 mRNAs in Figure 3B are statistically significant (p<0.05) in an unpaired t-test. This is especially so for ACT1, where the points + and – CHX essentially fall on the same curves. P-values for these differences need to provided in the figures or legends.

- Subsection “Slowing translation elongation protects transcripts against degradation”: Figure 3—figure supplement 6 is not the correct citation.

- Subsection “Slowing translation elongation protects transcripts against degradation”: The replicates are in Figure 3—figure supplement 2, not Figure 3—figure supplement 1; Are they citing Figure 3—figure supplement 1C? (This underscores the need to cite specific panels in the Figure supplements.

- Subsection “Slowing translation elongation protects transcripts against degradation” and Figure 3—figure supplement 3B: The stabilization of ACT1 is not convincing. As noted above, it appears that the experiments on total mRNA in Figure 3—figure supplement 1A-C were done on only one replicate.

- Subsection “Slowing translation elongation protects transcripts against degradation”: Why then are the results for total vs polyA(+) mRNAs essentially the same at 0.2μg/ml CHX?

- Subsection “Slowing translation elongation protects transcripts against degradation” and Figure 3D-G: As noted above, the results of a t-test show that the half-life differences conferred by 3AT are not significantly different for PDC1 and CIS3, but they claim that only one mRNA was unchanged without identifying it. Stipulate whether it's PDC1 or CIS3 that is judged to be unchanged.

- Subsection “Slowing translation elongation protects transcripts against degradation”: Indicate the number of Gly codons for the mRNAs in Figure 3E-G.

- Subsection “Slowing translation elongation protects transcripts against degradation” and Figure 3—figure supplement 3D: This statement is not likely to be true for ACT1, which appears to be unresponsive to 3AT in total RNA.

- Subsection “Inhibition of translation initiation destabilizes individual transcripts” and Figure 3I: As noted above, the results of a t-test show that the half-life differences conferred by HIP are not significantly different for ACT1 and RPL25. Destabilization of ACT1 and RPL25 mRNA appears to have occurred in total RNA (Figure 3—figure supplement 3E) but replicates would be needed if these results will be used to bolster the conclusions for these two mRNAs in polyA(+) RNA.

-Figure 3—figure supplement 5A-B: They claim that *S. pombe* cells were added for the polysome separation. Were these cells also treated with hippuristanol? If not, then presumably the reductions in polysomes conferred by HIP is even larger than indicated. This needs to be explained better in the legend. As noted above, the experiments in panels A-D appear to have been done only once, with no biological replicates. In addition, in Figure 3—figure supplement 5C-D: It's surprising that only RPL25 shows a shift to smaller polysomes, whereas ACT1 and CIS3, if anything, exhibit a shift to larger polysomes in the eIF4E/Gdown cells, which is not expected for a defect in initiation. In addition, the overall levels of the transcripts are greatly reduced here, but not in the HIP treated cells shown in panel B, despite similar effects of the two treatments on mRNA half-life shown in Figure 3I-J.

- Subsection “Translation elongation and initiation globally affect mRNA half-lives”: The median shifts in half-life are only 1-2 min on 3-AT treatment of HIP treatment in Figures 3H and 3K; and with CHX, it appears that the shift is restricted to the mRNAs with the longest half-lives. Given that mRNA half-lives vary over more than an order of magnitude, it's unclear that the changes in half-lives of 1-2 min conferred by substantially inhibiting initiation or elongation rates are adequate to explain the natural range in half-lives.

- Subsection “Inhibition of translation initiation induces processing bodies”: Stipulate that the marker for SGs is Pab1 (I presume) and cite the relevant panel of Figure 4—figure supplement 1. Explain better the dhh1 mutants analyzed in panel C.

Subsection “Inhibition of translation initiation induces processing bodies”: Explain better what "slowly decaying PP7 stem loops" means and indicate where they are located in the mRNA.

- Discussion section: The logic here is unclear. In Pgal shut-off experiments, cells are shifted to glucose to shut off new synthesis and the amounts of the remaining mRNAs are followed by Northerns. Growth in glucose is not stressful, but it could be that the carbon source shift per se alters the level or function of the decay machinery. I don't see how effects of GC-content on transcription would affect the measurements of half-lives by this approach.

- Discussion section: This conclusion needs to be elaborated as it is not self-evident.

Reviewer #2:

I agree with reviewer 1 that in an ideal world the authors would had employed an rRNA subtraction procedure such as ribo0, in addition to the total RNA analysis, as this would have allowed them to recover far more mRNA reads. Nonetheless, the data do argue strongly against the possibility that the discrepancies between the authors' mRNA half life data and previous studies was simply an artifact of looking at polyA selected mRNAs. This experiment and the expanded discussion thus address the major concern I had with the original submission.

Reviewer #3:

I find the manuscript to be significantly improved, and the authors have done an excellent job of addressing the first round of comments. I am now supportive of publication.

---

## [Author Response]

Summary:Chan et al. present an improved method for measuring the kinetics of mRNA synthesis and degradation that minimizes perturbation of cellular state and that accounts for various potential sources of non-biological variation. They apply this method to measure the half-lives of each type of mRNA in yeast and produce results that are strikingly different from those produced by earlier approaches. In particular, they find that average half-lives are substantially shorter than previously measured. Seeking to explain gene-by-gene variation in mRNA stability, the authors identify rates of translation initiation and optimal codon content as having similarly strong positive correlations with half-life. To distinguish between the more historical model in which the presence of ribosomes protects messages from decay machinery and a newer model in which slow elongation (e.g. of non-optimal codons) serves as a general trigger of decay, the authors measure mRNA stability after applying a variety of chemical and genetic methods of interfering with elongation or with initiation. They find that slow elongation stabilizes mRNA, while slow initiation destabilizes mRNA, favoring a model in which competition for access to the cap, rather than sensing of slow moving ribosomes, serves as the primary gatekeeper for decay pathways.Essential revisions:The most important issue to be addressed is whether your use of poly(A) selection has led you to measure the half-lives of only a fraction of mRNAs with poly(A) tails long enough for oligo-dT selection or, in the limit case, to measure rates of deadenylation only. It is necessary that you alter your technique to avoid poly(A) selection by using ribo0 depletion as an alternative approach and determine whether your major conclusions about mRNA half-lives and the relatively greater importance of translation initiation versus elongation rates in determining half-lives still hold.

To address this important issue, we have now performed decay experiments in the complete absence of any mRNA enrichment by analyzing total RNA samples. We opted for this approach –instead of ribo0- to avoid any potential bias that could be introduced by a selection step. Overall, these new measurements confirm our prior conclusions. First, using transcriptome-wide decay profiling we find that the transcriptome has almost identical average and median half-lives with very similar shaped distributions in the unselected and poly(A)-selected datasets supporting our finding that the transcriptome is shorter lived than most previous measurements have found. On an individual transcript level, we do observe differences between the two datasets. We show that this is in part due to lower sequencing coverage in the unselected samples and but also partly reveals true biological differences likely in steps downstream of deadenylation.

Second, we have gone back and reanalyzed all of the translational perturbation experiments in the absence of any mRNA enrichment. Again, these new results are entirely consistent with the polyA selected data, and if anything, we find more exaggerated effects, especially in the presence of the elongation inhibitors cycloheximide and sordarin (as was pointed out by reviewer 3).

Second, the reviewers were not convinced of the statistical significance of differences in half-lives of specific mRNAs being claimed in the Results, requiring more thorough documentation of replicates with the appropriate statistical analyses.

We have included all biological replicates and computed average half-lives with standard deviations and indicated changes in half-life that were statistically significant. We have also included P-values for all comparative cumulative histograms in the Results section to quantify the significance of the differences observed.

Third, because your most significant results at odds with the previous model positing that mRNA decay is triggered by stalled ribosomes containing an empty A-site came from analyzing turnover rates in gcn2 cells treated with 3AT, this approach should be conducted genome-wide to analyze the correlation (or lack thereof) between ribosome stalling at His codons and mRNA half-life.

As requested we performed a transcriptome-wide profile of mRNA half-lives in the presence and absence of 3AT. We find that there is significant stabilization of mRNAs across the transcriptome in 3AT treated cells, strengthening our prior conclusion. This analysis also suggests there is a thresholding effect where a minimal number of histidine and glycine codons must be present for a strong stabilization effect to be observed.

Fourth, for the experiments using hippuristanol or dominant-negative eIF4E/degron depletion of eIF4E to inhibit translation initiation, it should be verified that the translation initiation rate was reduced for the specific mRNAs whose half-lives are being measured.

To our knowledge translation initiation rates were never been directly measured in vivo. Instead we have therefore performed as suggested by reviewer 1, polysome fractionation in both hippuristanol-treated cells and in the double eIF4E/G mutant and analyzed how *ACT1, CIS3* and *RPL25* mRNAs partition in the polysome in treated and untreated conditions. We find that in the case of hippuristanol treatment, ribosomes are redistributed from heavy polysome fractions to the lighter fractions. This redistribution is also seen for specific mRNAs that were examined. In the case of the eIF4E/G double mutant, we find that overall polysome abundance is attenuated and this is once again mirrored for the specific transcripts.

Fifth, additional revisions of the text are required to reconcile your model with the results of previously published experiments on reporter mRNAs that provided strong evidence that codon optimality is a strong determinant of mRNA half-life, which would suggest that this mRNA feature has not merely co-evolved with translation initiation efficiency in the manner you suggest.

We have included a broader discussion of the possible sources of discrepancy between our observations and previous observations regarding codon optimality. This also includes a discussion of more recent work that has shown that altering codon usage in a model transcript often leads to increased GC content which in turn accelerates transcription which can falsely lead to perceived “stabilization” of a transcript due to higher steady state abundances.

Sixth, it seems necessary to address the alternative interpretation of your experiments on P-bodies raised by reviewer #1.

We have expanded our discussion of the site of mRNA decay in the cell and have tried to provide are more multifaceted and nuanced interpretation of these experiments. We try to make explicit that we view mRNA decay occurring not just in P-bodies and the polysome but we reason –also based on other recent publications on this topic- that neither of these sites are likely the primary sites of mRNA decay and that it is more likely that most mRNA turnover occurs in sub-microscopic decay mRNPs.

Reviewer #1:This paper describes an improved, non-invasive method for genome-wide measurements of mRNA half-lives in growing yeast cells. The results suggest that mRNA half-lives are between 3-fold and 26-fold shorter than determined previously by other approaches relying on metabolic labeling or inhibition of transcription, respectively. By examining the effects of inhibiting translation elongation or initiation on mRNA half-lives in various ways, that conclude that the rate of translation initiation is a more important determinant of mRNA half-life than is the rate of elongation, at odds with recent reports from the Coller and Green labs indicating that codon optimality is the principle determinant of mRNA stability of native mRNAs in budding yeast. They provide evidence that stalling ribosomes during elongation with the inhibitors cycloheximide or sordorin, or by starving for the amino acid histidine, tends to decrease rather than increase rates of mRNA turnover, presumably by having stalled ribosomes protect the mRNA from nucleolytic digestion, rather than inducing turnover via Dhh1 recognition of stalled ribosomes as recently concluded by Coller and Green. By contrast, an inhibitor of eIF4A or mutants that reduce eIF4F function and presumably produce a global decrease in the initiation rate lead to increased mRNA turnover. Thus, they propose that the rate of initiation is the key determinant of mRNA stability, with a reduction in initiation either exposing the mRNA cap to the decapping machinery or decreasing ribosome densities on coding regions. Based on the fact that inhibiting eIF4A with hippuristanol (HIP) induces Dhh1-dependent P-bodies, and that the abundance of a stem-loop reporter mRNA in P bodies decayed over time, they propose that inhibiting initiation sends mRNAs to P bodies and that this is, or can be, the site of their decay.- The differences in half-life claimed for ACT1 and CIS3 mRNAs in response to CHX treatment in Figure 3—figure supplement 1 and Figure 3B are unconvincing, and they have provided no half-life values for these experiments. The claim that RPS2, RPL25, CIS3 and OSW5 mRNAs exhibit slower turnover on 3AT treatment are similarly unconvincing.

As discussed in our response to point (2) above, we have now computed average half-lives with standard deviations for these experiments. This analysis revealed that *CIS3* is not significantly stabilized in the presence of 3AT, and we have modified the results to reflect this.

- The most significant results presented here that are at odds with the proposal of Coller/Green, that decay is triggered by stalled ribosomes containing an empty A-site, came from analyzing turnover rates in gcn2 cells treated with 3AT to induce histidine starvation and attendant stalling by elongating ribosomes at histidine codons genome-wide. The experiments using CHX or sordorin are not relevant because these drugs do not stall ribosomes with an empty A site. As such, in order to justify their claim that inhibiting elongation in this manner slows down rather than accelerates mRNA decay on a genome-wide level, it is necessary to obtain mRNA turnover data for the entire transcriptome using their technique in 3AT-treated gcn2 cells, rather than examining only a dozen mRNAs by RT-qPCR. In addition, because the previous ribosome profiling experiments using 3AT carried out in the Green lab, which they cite for the effect of 3AT in stalling ribosomes, employed a GCN2+ strain, it is necessary that they confirm that treating a gcn2 mutant with 3AT will cause a similar (or even greater) degree of pauses at histidine codons by elongating ribosomes, and to demonstrate a widespread lack of correlation between pausing at histidine codons measured by ribosome profiling and mRNA half-lives measured by their technique in 3AT-treated cells.

Please see our response to point (3) above.

- The authors have used treatment with hippuristanol (HIP), or overexpression of a dominant-negative eIF4E variant lacking cap-binding activity together with degron depletion of eIF4E, as alternative means to inhibit global translation initiation rates. However, they have relied only on cell growth assays to document these effects. It is necessary to show by a more direct assay that translation initiation is impaired for the specific mRNAs whose half-lives are being measured (ACT1, CIS3, and RPL25). The best approach would be to show that the average number of ribosomes per mRNA is reduced on these mRNAs by the treatments shown in Figure 3H-I by probing for these mRNAs across fractionated polysomes; the effects on abundance and average size of bulk polysomes will be obtained in parallel. Alternatively, they could show that the amounts of these mRNAs associated with eIF4G (RNA-IP) are reduced by the two treatments. Treatment with sordarin or CHX would be a key negative control for either approach.

Please see our response to point (4) above.

- In Figure 3—figure supplement 2C-D, there are unexplained increases in synthesis of RPL25 in response to CHX and sordorin that require comment.

We hesitate to propose some sort of ribosomal protein gene feedback mechanism as it is well outside the scope of this manuscript and have thus opted to not make such an addition.

- The interpretation of the results in Figure 4E, that the reporter mRNA is being degraded in P-bodies, seems to overlook the possibility that the reporter mRNA exchanges rapidly between P-bodies and other compartments, in which case it could be degraded elsewhere, e.g. on polysomes, and its abundance in P-bodies would merely indicate the overall cellular level of the transcript. Is there a way to eliminate this alternative?

Please see our response to point (6) above.

- The validity of this method for measuring mRNA half-lives depends on highly efficient incorporation of 4-thioU into all newly synthesized mRNAs, and also highly efficient separation of these mRNAs from the un-labeled mRNAs pre-existing before the 4-thioU pulse, as it is the decay of the later over time that is quantified to determine half-lives. Extracting decay rates from the data requires a theoretical model for the decay, which includes an "efficiency parameter". It is difficult to assess the validity of this model, and hence, the mRNA half-lives determined using it. This is worrisome because their measured half-lives are 3-fold shorter than those measured by other labs using non-invasive metabolic labeling, and there is no attention given to resolving these discrepancies. The authors have provided and extended technical supplement on the refinement of their technique, but it seems important to provide justification for the validity of the modeling and method for extracting turnover times in the main text of the paper.

We have extended our discussion of the justification of an efficiency parameter in the main text but reserve the full discussion to the extended technical supplement as this important but technical point would distract from the biological questions the manuscript is attempting to address. We would also point out that the gold standard for metabolic labeling to date which is 14C-adenosine incorporation agrees very well with our observations.

Reviewer #2:Chan et al. present an improved method for measuring the kinetics of mRNA synthesis and degradation that minimizes perturbation of cellular state and that accounts for various potential sources of non-biological variation. They apply this method to measure the half-lives of each type of mRNA in yeast and produce results that are strikingly different from those produced by earlier approaches. In particular, they find that average half-lives are substantially shorter than previously measured. Seeking to explain gene-by-gene variation in mRNA stability, the authors identify rates of translation initiation and optimal codon content as having similarly strong positive correlations with half-life. To distinguish between the more historical model in which the presence of ribosomes protects messages from decay machinery and a newer model in which slow elongation (e.g. of non-optimal codons) serves as a general trigger of decay, the authors measure mRNA stability after applying a variety of chemical and genetic methods of interfering with elongation or with initiation. They find that slow elongation stabilizes mRNA, while slow initiation destabilizes mRNA, favoring a model in which competition for access to the cap, rather than sensing of slow moving ribosomes, serves as the primary gatekeeper for decay pathways.This is important work that has been carefully designed and executed. The unexpectedly strong connection between codon optimality and mRNA decay has emerged as a major area of research over the last three years, and intense efforts are underway to understand the molecular mechanisms that underpin this connection. By calling into question one of the central assumptions of this new sub-field (namely, that slowing down elongation should destabilize mRNA), Chan et al., trigger a messy but necessary rethinking of our understanding of these phenomenon.1) Presnyak et al., 2015 argues that enriching for mRNA by polyA pulldown produces artificially depressed estimates of half-lives due to the existence of stable deadenylated messages. Why did the authors use polyA pulldown in their protocol? Could this effect contribute to the shorter half-lives they observe? (It is unclear exactly which datasets from this paper are the two Coller datasets in Figure 1—figure supplement 1G.)

Please see our response to point (1) above.

2) The authors argue that, given their results, the observed positive correlation between optimal codon content and half-life is likely to be due to co-evolution rather than direct causal connection. Several papers (PMIDs: 25768907, 27436874, 27641505, and others), however, have presented compelling evidence for a direct causal connection by showing that systematically deoptimizing the codon content of a reporter construct systematically reduces its stability. To some extent, the authors' results contradict the standard interpretation of these published experiments. The paper would be strengthened if the authors engaged with this contradiction head-on by more thoroughly discussing exactly which ideas in this field are and are not consistent with their new data.

Please see our response to point (5) above.

Reviewer #3:This manuscript addresses the relationship between steps in translation and mRNA degradation in yeast. The manuscript addresses an important and timely issue in the field: Is decay rate coupled to stalled ribosomes, or to poor translation initiation? This has become an issue because of two competing lines of work. First, previous work using reporter mRNAs showed that translation initiation was in competition with decay (both deadenylation and decapping). However, more recent work, using both genome wide analyses and reporter mRNAs has argued that inefficient translation elongation (as defined by non-optimal codons) is the predominant driver of mRNA decay rates. Thus, resolving this issue will be important for understanding mRNA decay control.The strength of this manuscript is that is uses genome wide analyses of mRNA decay rates both under normal conditions, and with perturbations that affect ribosome elongation rates or translation initiation rates. The main conclusion from the work is that slow ribosome movement actually stabilizes mRNAs, while decreasing translation initiation increases mRNA degradation. This is a useful contribution and worthy of consideration for publication in eLife, although there are substantive issues that should be addressed before publication.1) The biggest issue with the manuscript is that the procedure for mRNA decay rates involves an oligo(dT) selection step. Since many mRNAs in yeast exist at steady state as oligoadenylated or deadenylated mRNAs (see Herrick et al., 1990 MCB), this only selects for a substep of mRNAs (typically with A tails longer than 30). Thus, the approach is really only measuring the deadenylation rate of mRNAs and not overall decay. This is presumably why their decay rates are faster than what has been described previously using methods that do not rely on selection of A+ mRNA species. The best approach for the authors here would be to repeat the experiments using ribo0 RNA, which would allow them to access the full mRNA decay pathway. Alternatively, they could re-write the manuscript as simply studying deadenylation, but this would diminish the impact of the work.

Please see our response to point (1) above.

2) In Figure 3B, the differences in decay curves are slight and would need to be supported by error bars to convince the reader these are meaningful. In fact, the data might be more convincing if decay rate was measured with ribo0 RNA since previous work (Beelman and Parker, 1995 and others) had shown cycloheximide primarily inhibits decapping with little effect on deadenylation rate.

Please see our response to point (2) above.

[Editors' note: further revisions were requested prior to acceptance, as described below.]

All three reviewers agreed that the paper has been significantly improved by the new experiments that were carried out, as well as the revisions of text; and your efforts to address all of the major criticisms of the first version were greatly appreciated. In particular, the new genome-wide data obtained from analyzing total RNA generally supports the conclusions reached from poly(A) selection, which resolves the most important criticism of the original version of the paper.

Thank you and we agree that the new genome-wide data have strengthened our manuscript.

There are, however, some remaining issues that require your attention, raised by reviewer #1, including the lack of statistical significance of small differences in mRNA half-lives for certain mRNAs, and the apparent lack of any replicates for the new half-life measurements conducted with total RNA. There is also an objection to your interpretation of the new results presented to establish a shift to smaller polysomes of mRNAs in response to down-regulating eIF4E/4G; and a concern that these experiments might have been conducted only once without replication. It is felt that these shortcomings detract from the overall scientific quality of the work and do not satisfy the journal's rigorous standards for replicate measurements and statistical analyses of data. Finally, you have been asked to consider whether the magnitude of the observed effects of reducing initiation or elongation in decreasing or increasing mRNA stabilities, respectively, are substantial enough to account for the order-of-magnitude-range of mRNA stabilities observed in wild-type cells.

In response to this, we have subjected our individual transcript measurements to a more rigorous statistical analysis and we have found that the majority of our measurements support a model where competition with translation initiation determines the stability of an mRNA. None of our results refute this model. Furthermore, we have found no evidence for the stalled ribosome leading to mRNA decay model. In light of this analysis and the lack of ambiguity in our findings, we would argue that further biological replicates for the same experiments using total RNA (rather than polyA selected mRNA) would only serve to delay publication of this manuscript.

Furthermore, we have removed the polysome analysis from the main paper and included it only as a figure in response to reviewers as this was an experiment that was originally requested by comments of reviewer 1. This experiment only serves to bolster the assertion that translation is attenuated in hippuristanol treated cells and in eIF4E/G^down^ mutant cells, which we have already demonstrated clear growth defects for. Moreover, repeating the hippuristanol experiment is not a simple matter as this drug is not commercially available. These experiments require large amount of drug and the small amount that we have was a generous gift of our collaborator, Junichi Tanaka, who unfortunately could not provide us with more hippuristanol at this point.

Lastly, given the results of our statistical analysis, we decided against softening our conclusions. However, we have included an extensive discussion where we attempt to account for the differences between the Coller group conclusions and our findings.

Reviewer #1:It's unfortunate that they chose not to use ribo0 to deplete rRNA, as only a few percent of the total mRNA reads they obtained in the new experiments come from mRNA, and they claim that the relatively weak correlation between mRNA half-lives for total vs polyA selected mRNA is attributable in part to the low read depth for total RNA. Nevertheless, they find the same very short half-life for bulk total mRNA observed for bulk polyA(+) mRNA, which supports the validity of their measurements using polyA selected mRNAs.

We opted to analyze total RNA to avoid any biases that are introduced by RNA selection methods, and we concur that our new analysis fully supports the validity of our initial measurements.

There are remaining issues however concerning the changes in half-lives of specific mRNAs in response to different drugs or genetic manipulations. They did no significance testing on these measurements but based on the mean values and standard deviations provided for 3 biological replicates, my own application of a standard t-test indicates that the changes shown in Figure 3B for low-level CHX treatment are not statistically significant (p>0.05) for any of the three mRNAs (ACT1, CIS3, RPL25). This is also the case for PDC1 and CIS3 in response to 3AT in Figures 3D and 3F (one of which was claimed to show a difference, but it wasn't indicated which one), for ACT1 and RPL25 in response to HIP in Figure 3I, and for ACT1 in response to the eIF4E/4G_down mutations in Figure 3J. It could be argued that the new measurements conducted on analysis of total RNA shown in Figure 3—figure supplement 3 can be used to bolster their conclusions, but these measurements were done for only a single biological replicate; moreover, the findings for total RNA for ACT1 again seem to show no difference in half-life for +/- CHX (panel B), and none as well for ACT1 and RPL25 in response to 3AT (panel D). (CIS3 is stabilized in panel D, but it's unclear whether CIS3 was judged to be stabilized in the polyA(+) measurements, as noted above.) Two additional replicates are needed for the total RNA measurements, allowing them to provide mean and SD values. In addition, they need to indicate which differences in Figure 3 and Figure 3—figure supplement 3 (once more replicates are conducted for the latter), have a P-value of <0.05 in the appropriate significance test.

We thank the reviewer for encouraging us to consider our data in a more rigorous statistical framework. Since all drug treatment experiments were performed by collecting a treated and mock treated sample at the same time, we have used a paired t-test and to allow for hypothesis testing of the two models of decay determination (ribosome stalling vs translation factor protection), we have sampled from a one-tailed distribution. We have included all P-values in a revised Figure 3. In summary, 21 of 27 measurements (lowCHX: CIS3, RPL25; SOR: ACT1, CIS3, RPL25; 3AT: CYS4, DED1, CIS3, RPL25, AIM7, AIM13, RPS2, NPA3, PAM16, RPC11, VMA21; HIP: ACT1, CIS3, RPL25; eIF4E/G: ACT1, CIS3) have p-values less than 0.05 in support of the prediction of the translation factor protection model. 6 of the measurements (lowCHX: ACT1; 3AT: ACT1, PDC1, OSW5, TPI1; eIF4E/G: RPL25) have p-values greater than 0.05 and therefore cannot be said to be statistically significant. In contrast, 0 of the 21 measurements testing the stalled ribosome model have p-values less than 0.05 (we excluded the HIP and eIF4E/G measurements from this analysis since the stalled ribosome model does not make a prediction for these experiments). Thus, we conclude that the majority of our data support the translation-factor protection model and none of the data we have collected support the stalled ribosome-triggered decay model. We have included the additional statistical analyses in the text and Figure of a revised manuscript.

Furthermore, we would like to point out that these small-scale experiments are significantly bolstered by the transcriptome-wide analyses that are presented in the manuscript and the global trends are completely in line with our small-scale experiments. Moreover, the effect we have observed with cycloheximide has been previously reported (Beelman and Parker, 1994). In addition, the experiments measuring mRNA stability in the absence of polyA selection were performed in response to reviewers’ comments and were only included as a supplemental figure. Given the statistical analysis described above and given that none of the trends change (and if anything, are further exacerbated for certain experiments such as the 50 μg/mL cycloheximide and sordarin experiments), we would argue that further biological replicates would not add significant value to the manuscript and would only serve to further delay the publication of this work.

There are also issues with the new experiments added to show that the initiation rates of particular mRNAs are reduced in response to HIP or eIF4E/4G_down. It appears that the experiments were done only once, as there is no mention of data from replicates. Also, the data in panel D of Figure 3—figure supplement 5 for all three mRNAs are quite unexpected, as the mRNA levels are greatly reduced, and the ACT1 and CIS3 mRNAs actually do not shift to smaller polysomes, in response to eIF4E/4G_down. By comparison, the HIP treatment in panel B shows no substantial change in overall mRNA level despite similar decreases in half-life for these mRNAs in the two different treatments and does exhibit the expected shift of all 3 mRNAs to smaller polysomes. It needs to be shown that the results in panels B and D are reproducible in at least one replicate experiment, and the unexpected data for the eIF4E/4G_down experiment need to be discussed.

We would like to respectfully disagree that the lower overall mRNA levels in eIF4E/G are unexpected. The effect of attenuated eIF4A is likely not a complete block in initiation as cap recognition is still possible and the importance of eIF4A in efficient translation is likely transcript specific as suggested by papers of the Ingolia, Fanidi and Hinnebusch groups. In limiting hippuristanol where eIF4A is partially inhibited, one would expect that without full helicase activity, ribosomes would likely kinetically pile up during the scanning phase prior to AUG recognition and a shift from heavier polysomes to lighter polysomes would be expected. This is indeed what is observed. However, in the eIF4E/G mutant, both the cap binding subunit as well as the scaffolding subunit are attenuated but all other translation initiation machinery is intact and functional and thus any ribosome that loads onto an mRNA should behave normally. Given the critical role of the pioneer translation initiation round for sustained translation cycles (reviewed by Maquat, Tarn and Isken, 2011), general mRNA attenuation in the polysome is a logical outcome.

The original purpose of this figure was to address a reviewer’s comment that an independent means of verifying translation attenuation other than reduced growth was required. The data presented have clearly demonstrated that translation is indeed defective.

With respect to biological replication, this is currently not feasible as this experiment requires a large amount of hippuristanol, which had received only a limited amount from a generous collaborator. Moreover, as we performed this experiment to address a reviewer’s concerns we decided to exclude this figure from the main manuscript and now only include it in the response to reviewer’s comments. We feel this is appropriate as the in depth discussion of the results of this experiment are not at all the focus of this paper and would only serve to distract and dilute the main points of the manuscript.

The median shifts in half-life for all mRNAs shown in Figure 3H and 3K are only 1-2 min on 3-AT or HIP treatment; and with CHX treatment, it appears that the shift is restricted to the mRNAs with the longest half-lives. It seems possible therefore that the effects of reduced initiation or elongation in decreasing or increasing mRNA stability, respectively, are not really strong enough to explain the order of magnitude range of mRNA stabilities observed in wild-type cells. Perhaps the authors should consider softening statements about initiation being the primary determinant of mRNA stability and instead identify it as one important determinant. Is it possible that on some mRNAs, the presence of suboptimal codons is equally important, with ribosomes containing empty A sites being recognized by Dhh1 in the manner suggested by Coller and Green? After all, two out of 12 mRNAs were found to decay faster, not slower, in the presence of 3AT; and presumably there are many other mRNAs behaving like these two mRNAs, in a manner consistent with the Coller/Green Dhh1 model.

As was previously requested by a reviewer, we have presented the significance values for the transcriptome-wide data and the p-values indicate significance. Moreover, in light of the statistical analysis of the single decay plots which revealed that the large majority of our measurements are in significant support of the translation-factor protection model, none of our unbiased experiments argue against the translation-factor protection model and importantly, none of our measurements came out in support of the ribosome-triggered decay model, we feel that it would be inappropriate to soften our claims or suggest that ribosome-stalling plays a role in general mRNA decay rate determination. We have suggested in the discussion that this model may make more sense in the stress conditions that were used to measure mRNA stability of engineered transcripts or for select individual transcripts.

We would also like to reiterate that we are not proposing a new model in this manuscript. Rather, we are testing in an unbiased manner two prevailing theories of how mRNA stability is determined. We have found that our data support the model proposed by Parker and others in the 1990s which is based on data gathered from multiple types of experiments where translation initiation is perturbed either in cis or in trans and mRNA stability was found to be decreased (see Beelman and Parker, 1994, Schwartz and Parker, 1999 and LeGrandeur and Parker, 1999). Furthermore, our data are the first to contradict a stalled ribosome-triggered decay model. Studies using mutant tRNAs to induce ribosome pausing during elongation have shown that this leads to stabilization of mRNAs (Peltz, Donahue and Jacobson, 1992 and Zuk, Belk and Jacobson, 1999). Moreover, we have presented a fair discussion attempting to resolve the differences between our experiments and those of Coller/Green.

With respect to magnitude, we intentionally chose doses of drug treatments that had modest effects on cell physiology. Indeed, when strong doses are employed as in the case of high doses of cycloheximide, we and others previously have reported dramatic effects on mRNA stabilization. However, we wished to measure half-lives in cells that were not completely shut down and therefore chose drug doses that were well below lethal levels. We would contrast this with the methods of Coller/Green where artificial, model mRNAs are codon deoptimized well below any observed codon optimality found in the transcriptome.

- Subsection “An improved non-invasive metabolic labeling protocol reveals that the yeast transcriptome is highly unstable”: Perhaps step 7 is intended?

We have updated the text to resolve this ambiguity.

- Subsection “An improved non-invasive metabolic labeling protocol reveals that the yeast transcriptome is highly unstable”: Indicate the medium used, e.g. nutrient-rich medium with glucose as carbon source?

We provide a complete formulation of the media used in the Materials and methods section. To do so in the body of the text would be disruptive.

- Figure 1—figure supplement 1 has 9 panels. Relevant panels should be cited individually in the text-otherwise the reader has to inspect the entire figure without the guidance of text to find the supporting results being cited. This should be fixed throughout the manuscript.

We agree and have fixed this issue.

- Subsection “An improved non-invasive metabolic labeling protocol reveals that the yeast transcriptome is highly unstable”: The pool of polyadenylated mRNAs should be designated polyA(+) mRNA rather than polyA-mRNA, which could be interpreted as polyA(-).

We thank the reviewer for the feedback and have changed this.

- Figure 1—figure supplement 2B: Indicate whether the correlation value of 0.44 is r or r2.

The value is presented as Pearson correlation, which is by definition r. We have now indicated that the value is r in the figure.

- Subsection “Slowing translation elongation protects transcripts against degradation”: Weinberg and Bartel showed that mRNA abundance is strongly correlated with translation efficiency, so the correlation with abundance could also be reflect the correlation of half-life with TE.

We agree that all of these parameters correlate with one another. We present one possible explanation of one of these correlations, but we do not draw any conclusions as this is merely correlation and causation cannot be implied.

- Was Figure 2—figure supplement 1 cited in Results section?

It is only cited in the Discussion section. We did not include this in the Results section as this translation initiation rate dataset was derived using a kinetic model and is not a result of direct initiation rate measurements. Thus we did not deem it fair to include in the analysis in the main Figure 2.

- Subsection “Slowing translation elongation protects transcripts against degradation” and Figure 3—figure supplement 1A and Figure 3B: It needs to be stated in Results section or at least Figure 3 legends that the data for the specific mRNAs was obtained by RT-qPCR quantification of mRNAs obtained from the mRNA stability profiling technique outlined in Figure 1. There are no error bars for the 50μg/ml CHX experiment in Figure 3—figure supplement 1A, indicating the apparent lack of replicates. As noted above, assuming that the error bars in Figure 3B (which were not defined in the legend) are the SD values for n=3, none of the differences in half-life for the 3 mRNAs in Figure 3B are statistically significant (p<0.05) in an unpaired t-test. This is especially so for ACT1, where the points + and – CHX essentially fall on the same curves. P-values for these differences need to be provided in the figures or legends.

This has been addressed as discussed above.

- Subsection “Slowing translation elongation protects transcripts against degradation”: Figure 3—figure supplement 6 is not the correct citation.

The citation refers to the titration of cycloheximide to a sub-lethal dose, which is Figure 3—figure supplement 6H. The new in text citation of subpanels in the supplemental figures hopefully clarifies this.

- Subsection “Slowing translation elongation protects transcripts against degradation”: The replicates are in Figure 3—figure supplement 2, not Figure 3—figure supplement 1; Are they citing Figure 3—figure supplement 1C? (This underscores the need to cite specific panels in the Figure supplements.

As stated above, we have fixed this issue.

- Subsection “Slowing translation elongation protects transcripts against degradation” and Figure 3—figure supplement 3B: The stabilization of ACT1 is not convincing. As noted above, it appears that the experiments on total mRNA in Figure 3—figure supplement 1A-C were done on only one replicate.

We have addressed and discussed this issue above.

- Subsection “Slowing translation elongation protects transcripts against degradation”: Why then are the results for total vs polyA(+) mRNAs essentially the same at 0.2μg/ml CHX?

Thank you, we have clarified the language.

- Subsection “Slowing translation elongation protects transcripts against degradation” and Figure 3D-G: As noted above, the results of a t-test show that the half-life differences conferred by 3AT are not significantly different for PDC1 and CIS3, but they claim that only one mRNA was unchanged without identifying it. Stipulate whether it's PDC1 or CIS3 that is judged to be unchanged.

We have addressed and discussed this issue above.

- Subsection “Slowing translation elongation protects transcripts against degradation”: Indicate the number of Gly codons for the mRNAs in Figure 3E-G.

We have included glycine and histidine amino acid contents for all transcripts in Figure 3E-G.

- Subsection “Slowing translation elongation protects transcripts against degradation” and Figure 3—figure supplement 3D: This statement is not likely to be true for ACT1, which appears to be unresponsive to 3AT in total RNA.

We did not find that using unselected mRNA altered the stability measurement for ACT1. Therefore, the statement that the “results were recapitulated” seems valid.

- Subsection “Inhibition of translation initiation destabilizes individual transcripts” and Figure 3I: As noted above, the results of a t-test show that the half-life differences conferred by HIP are not significantly different for ACT1 and RPL25. Destabilization of ACT1 and RPL25 mRNA appears to have occurred in total RNA (Figure 3—figure supplement 3E) but replicates would be needed if these results will be used to bolster the conclusions for these two mRNAs in polyA(+) RNA.

We have addressed and discussed this issue above.

- Figure 3—figure supplement 5A-B: They claim that S. pombe cells were added for the polysome separation. Were these cells also treated with hippuristanol? If not, then presumably the reductions in polysomes conferred by HIP is even larger than indicated. This needs to be explained better in the legend. As noted above, the experiments in panels A-D appear to have been done only once, with no biological replicates. In addition, in Figure 3—figure supplement 5C-D: It's surprising that only RPL25 shows a shift to smaller polysomes, whereas ACT1 and CIS3, if anything, exhibit a shift to larger polysomes in the eIF4E/Gdown cells, which is not expected for a defect in initiation. In addition, the overall levels of the transcripts are greatly reduced here, but not in the HIP treated cells shown in panel B, despite similar effects of the two treatments on mRNA half-life shown in Figure 3I-J.

We have addressed and discussed this issue above.

- Subsection “Translation elongation and initiation globally affect mRNA half-lives”: The median shifts in half-life are only 1-2 min on 3-AT treatment of HIP treatment in Figures 3H and 3K; and with CHX, it appears that the shift is restricted to the mRNAs with the longest half-lives. Given that mRNA half-lives vary over more than an order of magnitude, it's unclear that the changes in half-lives of 1-2 min conferred by substantially inhibiting initiation or elongation rates are adequate to explain the natural range in half-lives.

We have addressed and discussed this issue above.

- Subsection “Inhibition of translation initiation induces processing bodies”: Stipulate that the marker for SGs is Pab1 (I presume) and cite the relevant panel of Figure 4—figure supplement 1. Explain better the dhh1 mutants analyzed in panel C.

The marker for SGs is indicated in the figure. We have extended the explanation of the *dhh1* mutants in the text.

Subsection “Inhibition of translation initiation induces processing bodies”: Explain better what "slowly decaying PP7 stem loops" means and indicate where they are located in the mRNA.

We tried to explain this better in the revised text.

- Discussion section: The logic here is unclear. In Pgal shut-off experiments, cells are shifted to glucose to shut off new synthesis and the amounts of the remaining mRNAs are followed by Northerns. Growth in glucose is not stressful, but it could be that the carbon source shift per se alters the level or function of the decay machinery. I don't see how effects of GC-content on transcription would affect the measurements of half-lives by this approach.

Growth is glucose is not stressful but an acute shift from any carbon source to another will lead to remodeling of the gene expression profile brought upon by growth regulatory and nutrient sensing pathways. The GC-content effect on transcription is only relevant when stability is inferred from steady state abundance measurements, not direct kinetic measurements. This is more directly examined in Zhou et al.’s paper, “Codon usage is an important determinant of gene expression levels largely through its effects on transcription”. This is not directly referencing the Coller/Green methods. We have clarified this point in the text.

- Discussion section: This conclusion needs to be elaborated as it is not self-evident.

We have elaborated on this point in the text.

Reviewer #2:I agree with reviewer 1 that in an ideal world the authors would had employed an rRNA subtraction procedure such as ribo0, in addition to the total RNA analysis, as this would have allowed them to recover far more mRNA reads. Nonetheless, the data do argue strongly against the possibility that the discrepancies between the authors' mRNA half-life data and previous studies was simply an artifact of looking at polyA selected mRNAs. This experiment and the expanded discussion thus address the major concern I had with the original submission.

Thank you and we concur that the analysis of total RNA further bolsters our original conclusions.